# LIPG-mediated regulation of lipid deposition and proliferation in goat intramuscular preadipocytes involves the PPARα signaling pathway

Yinggui Wang[1], Lian Huang[1]*, JiangJiang Zhu[1,2]*, Wenyang Zhang[2], Yinmei Tang[1], Changheng Yang[1], Yaqiu Lin[1,2], Yong Wang[1,2], Hua Xiang[1,2]

**1** Qinghai-Tibetan Plateau Animal Genetic Resource Reservation and Utilization Key Laboratory of Sichuan Province, Southwest Minzu University, Chengdu, China, **2** Key Laboratory of Qinghai-Tibetan Plateau Animal Genetic Resource Reservation and Utilization (Southwest Minzu University), Ministry of Education, Chengdu, China

* 80300227@swun.edu.cn (LH); zhujiang4656@swun.edu.cn (JJZ)

## Abstract

Endothelial lipase (LIPG), a member of the triglyceride lipase family, plays an essential role in human diseases and lipid metabolism. However, its function in goat intramuscular fat (IMF) deposition remains unclear. In this study, we investigated the role of the *LIPG* gene in IMF deposition by knocking down and overexpressing it in goat intramuscular preadipocytes. We successfully cloned the full-length *LIPG* gene, which spans 2,131 bp, including a 94 bp 5' untranslated region (5'UTR), a 1,503 bp coding sequence (CDS), and a 534 bp 3' untranslated region (3'UTR). Tissue expression profiles showed that *LIPG* is expressed in the heart, liver, spleen, Kidney, longest dorsal muscle, and small intestine tissues of goats. *LIPG* knockdown significantly inhibited both the proliferation of intramuscular preadipocytes and lipid deposition. Moreover, *LIPG* knockdown markedly decreased mRNA expression of *FASN, LPL, CPT1A, CPT1B, FABP3*, while increasing the mRNA expression of *ATGL, ACOX1, FADS1,* and *ELOVL6*. These findings were further corroborated through *LIPG* overexpression experiments. Using RNA sequencing (RNA-seq), we identified 1695 differentially expressed genes (DEGs) between the negative control (NC) and *LIPG* knockdown (Si-LIPG) groups, with KEGG pathway analysis revealing significant enrichment in the PPAR signaling pathway. Additionally, *LIPG* knockdown significantly upregulated the expression of both mRNA and protein levels of PPARα. The PPARα agonist WY14643 was able to reverse the enhanced lipid deposition induced by *LIPG* overexpression. In conclusion, our study highlights a key role for LIPG in the regulation of goat intramuscular preadipocyte proliferation and lipid deposition, potentially through the PPARα signaling pathway. These findings provide new insights into the regulatory mechanisms governing IMF deposition and suggest potential strategies for improving goat meat quality.

**Data availability statement:** All relevant data are within the manuscript and its Supporting information files.

**Funding:** JiangJiang Zhu was supported by Sichuan Science and Technology Program (2021YFYZ0003); JiangJiang Zhu was supported by Zhejiang Science Technology Program (2022C04017); Hua Xiang was supported by Sichuan Science and Technology Program (2022NSFC0082); Yinggui Wang was supported by Graduate innovative scientific research project of Southwest Minzu University (ZD2022726); The funders had no role in study design, data collection and analysis, decision to publish, or preparation of the manuscript.

**Competing interests:** The authors declare that they have no known competing financial interests or personal relationships that could have appeared to influence the work reported in this paper.

## Introduction

In addition to being found in adipose tissues, adipocytes are also present in subcutaneous, mesenteric connective tissues, visceral, and intermuscular areas [1], playing crucial roles in lipid metabolism, energy regulation, and temperature maintenance [2–4]. Unlike in humans, where adipocytes are mainly associated with metabolic diseases such as insulin resistance, type 2 diabetes mellitus, and aging [5], intramuscular adipocytes—also referred to as intramuscular fat (IMF)—have been shown to positively correlated with meat juiciness and flavor in cattle [6], pigs [7], sheep [8] and goats [9]. Adipocytes have been widely used to investigate lipid metabolism regulation, and a range of genes have been shown to be involved in the lipid formation process [10]. However, the mechanisms underlying IMF deposition in goats remain poorly understood.

Endothelial Lipase (EL), also known as Lipase G (LIPG), is a lipase that belongs to triglyceride (TAG) lipase family, which also includes lipoprotein lipase (LPL) and hepatic lipase (HL) [11–13]. LIPG hydrolyzes high-density lipoprotein (HDL) and phospholipids (PL) to release lysophospholipids (LysoPL) and fatty acids (FAs) [14,15]. These FAs are then taken up and used for lipid synthesis, metabolism [16–18], cell composition, cytokine expression, and energy supply [19]. In humans, LIPG has been linked to metabolic syndrome [20], including inflammation, excessive obesity, atherosclerosis [21], hypertension [22], depression [23], leukemia [24], cancer (stomach cancer, lung cancer [25], breast cancer, ovarian cancer, colon cancer, cervical cancer, cervical cancer) [19,26], and other diseases due to its role in lipid metabolism and cell proliferation. In sheep, LIPG was identified as a major differential gene between fat-tailed and thin-trailed sheep with different lipid storage patterns [27].

Despite these insights, little is known about the role of LIPG in the regulation of goat intramuscular preadipocyte proliferation and lipid accumulation. In this study, we cloned LIPG coding sequences from goats and examined its role in cell proliferation and lipid deposition by knockdown and overexpression technology. Additionally, we demonstrated that the PPARα signaling pathway is involved in the regulatory process through RNA-seq and a rescue experiment. These findings provide new insights into the lipid metabolism regulation network and offer a theoretical basis for improving goat meat quality through genetic breeding techniques.

## Materials and methods

### Ethics statement

All experimental exercises were reviewed and approved by the Institutional Animal Care and Use Committee, Southwest Minzu University (Chengdu). The experimental animal certification number was SYXK-2020-1-20.

### Collection of tissue samples and isolation of goat intramuscular preadipocytes

Three 12-month-old Jianzhou goat were randomly selected from Sichuan Jianyang Dageda Animal Husbandry. Following euthanasia, the heart, liver, spleen, kidney, longest dorsal muscle, subcutaneous fat, and small intestine were promptly collected, washed with DEPC-treated water, and immediately frozen in liquid nitrogen for storage.

Goat primary intramuscular preadipocytes were isolated using the previously described method [28,29]. Briefly, longissimus dorsi tissues were collected from three-day-old Jianzhou goats after euthanized. The tissues were washed three times with PBS containing 1% penicillin/streptomycin and then were cut into 1 mm³ cubes using ophthalmic scissors. The samples were then digested with type II collagenase 0.2% type II collagenase (1 ml per gram of tissue)

for 1.5 h at 37°C with gentle shaking, and the digestion was terminated by adding an equal amount of complete medium (DMEM/F12 (Gibco, Beijing, China), 10% FBS (Gemin, Beijing, China), 1% Penicillin-Streptomycin Solution). The samples were then filtered using sterile gauze and a 75-μm cell strainer, and the cells were then centrifuged at 2000 rpm for 5 min. Following discarding the supernatant, and cells were resuspended by complete medium and standing for 5 min. The cells were then seeded into 25 cm² culture flasks and cultured under 5% $CO_2$ at 37 °C. Previous studies have shown that preadipocytes were adhere at 0-2 h and myogenic cells at 2-74 h [30]. Intramuscular preadipocytes were isolated by removing nonadherent cells from the supernatant after 2 h. The culture medium was changed every two days until the cell density reached 80%. Then the cells were passed to third generation and inoculated into 6-well plates.

## Gene cloning, sequence analysis and construction of phylogenetic trees

The cloning primers (S1 Table) were designed based on the predicted sequence of *LIPG* on Genbank (Genbank registry number: XM_005697191.3). The LIPG gene was cloned from goat subcutaneous adipocytes using the following primers: LIPG-F: CTTGGACCGCTGGAAAC, LIPG-F: CTTGGACCGCTGGAAAC. The sequences obtained by cloning were analyzed by DNAman 8 for sequence analysis. Phylogenetic trees were constructed by building MEGA 5.05.

## Vectors construction and chemical synthesis of siRNA

The CDS region of the LIPG gene was ligated into the pcDNA3.1 vector linearized by *EcoR* I and *xho* I, named OE-LIPG. PcDNA3.1 (+) plasmid was chosen as a negative control (OE-NC). The specific interfering siRNAs (siRNA-1(GCAGGAAGAACCGUUGUAATT), siRNA-2 (CUUGAGAACACCCUUAUAUTT)) were designed and synthesized based on *LIPG* gene by Shanghai GenePharma. Negative control (NC) was provided by GenePharma (UUCUCCGAACGUGUCACGUTT).

## Transfection, treatment and induced differentiation of goat intramuscular preadipocytes

The well-being of intramuscular preadipocytes were inoculated into 6-well plates, and transfection was initiated when cell confluence reached 70-80%. Each well of the 6-well plate was transfected with 1ug of plasmid or 120 μM siRNA, respectively. The transfection process was performed according to the instructions of Lipofectamine™ 3000 (Thermo Fisher Scientific). After the end of transfection, the cell culture medium was replaced with induction medium (MEM/F12, 10% FBS, 1% Penicillin-Streptomycin Solution, 50 μmol/L oleic acid (Sigma, Tokyo, Japan)) [31]. After 48 hours, these cells were utilized for subsequent analyses including Oil Red O staining, BODIPY staining, triglyceride assay, EDU staining, and qPCR.

## Treatment of orlistat

Transfected cells were treated with 20 μM orlistat (dissolved in DMSO) and cultured for an additional 48 h.

## Oil Red O staining and BODIPY staining

Oil Red O staining and BODIPY staining were referred to as the previously reported methods [32]. Briefly, cells in 6-well plates were washed three times with PBS and fixed with 4% formaldehyde at room temperature for 30 min. During the Oil Red O staining process, the Oil

Red O working solution (the mixture of 3 ml Oil Red 5 g/L dissolved in isopropanol and 2 ml of ddH$_2$O) was added to the cells and incubated for 20 min at room temperature. The cells were cleaned with PBS and photographed with microscope. Finally, 1 ml of isopropanol was added to each well of the 6-well plate and the absorbance value at 510 nm was measured using a spectrophotometer (Thermo Fisher Scientific, Shanghai, China).

For BODIPY staining, cells were incubated with BODIPY staining solution (1 μg/mL of BODIPY™ 493/503 (Thermo Fisher Scientific, D3922)) for 30 min in the dark. Cells were then washed three times with PBS and stained with DAPI solution (1 μg/mL of DAPI (Solarbio, C0060)) for 10 min. Finally, the cells were photographed by inverted fluorescence microscopy, and measured the fluorescence intensity and cell number by Image-Pro Plus 6.0.

## Triglyceride assay

Intracellular triglyceride content was measured by a Tissue Triglyceride (TAG) Content Assay Kit (Applygen, E1013), following the instructions. Briefly, the cells were washed three times with PBS, and treated with 200 μl of cell lysis on ice for 10 min. A mixture of 10 μl of lysate and 190 μl of working solution was incubated at 37°C for 15 min and the absorbance value at 550 nm was detected. The total protein was measured by BCA protein assay kit (Boster, Wuhan, China) for triglyceride standardization.

## EDU staining

The BeyoClick™ EDU-488 Cell Proliferation assay kit (Beyotime Biotechnology, C0071S) was used for EDU staining following the manufacturer's protocol. In summary, the cells were added to EDU solution preheated at 37 °C (final concentration of 10 μM) and continued to culture for 2 h. Next, cells were washed 3 times using PBS and were fixed by adding 1 ml of 4% paraformaldehyde for 15 min at room temperature. At the end of fixation, the cells were washed three times using washing solution (PBS with 3% BSA) and were permeabilized using permeabilization solution (PBS with 0.3% Triton X-100) at room temperature for 15 min. Then, the cells were washed twice with washing solution and incubated with 1 ml click working solution for 30 min at room temperature. Finally, the nuclei were stained using 1× Hoechst. Cells were photographed using an inverted fluorescence microscope and the photographs were quantitatively analyzed using Image-Pro Plus 6.0.

## Total RNA extraction and quantitative real-time PCR (RT-qPCR)

Total RNA from goat tissue or intramuscular preadipocytes was extracted using RNAiso Plus (Takara, 9109). The concentration and purity of RNA were measured using a Nanodrop 2000 (Thermo Fisher Scientific, Beijing, China), and the absorbance ratios (260/280 nm) of all RNAs were between 1.8 and 2.0. Reverse transcription of 1 μg total RNA to cDNA using the Reverse Transcription Kit (Vazyme, R323-01, Nanjing, China). Real-Time PCR was carried out with a Bio-Rad CFX96 PCR System using Taq Pro Universal SYBR qPCR Master Mix (Vazyme, Q712-02, Nanjing, China) and gene-specific primers (S1 Table). Ubiquitously expressed prefoldin like chaperone (*UXT*) was used as an internal reference gene and the relative expression was calculated using the $2^{-\Delta\Delta Ct}$ methods.

## RNA sequencing (RNA-seq)

Goat intramuscular preadipocytes were cultured in 6-well plates, transfected and induced for 48 h. Sterile PBS was used to wash the cells three times, 1 ml Trizol was added, and the samples were sent to Hangzhou LcbioTechnologies for transcriptome sequencing. DEsq2 was used to screen differentially expressed genes (DEGs) analysis. Gene Ontology (GO) and Kyoto

encyclopedia of genes and genomes (KEGG) analysis were analyzed using Sanger box (http://sangerbox.com/home.html). Gene Set Enrichment Analysis (GSEA) was conducted using Omicshare (https://www.omicshare.com/). Data were deposited in Gene Expression Omnibus database (GEO) (https://www.ncbi.nlm.nih.gov/geo/) and Sequence Read Archive (SRA) (https://www.ncbi.nlm.nih.gov/sra/) with accession numbers GSE234932 and PRJNA946935.

## Western blot

Total protein was harvested from goat intramuscular preadipocytes using Western and IP cell lysates (Beyotime Biotechnology, P0013, Shanghai, China) containing protease inhibitor mixture (Beyotime Biotechnology, P1008, Shanghai, China). PPARα antibody (Protein tech, 66826-1, Wuhan, China) and LIPG antibody (Protein tech, 67434-1, Wuhan, China) were used as primary antibody. β-actin (Boster, BM0627, Wuhan, China) was selected as an internal reference protein. Goat anti-mouse IgG-HRP (Boster, BA1050 Wuhan, China) and Goat anti-rabbit IgG-HRP (Boster, BA1054, Wuhan, China) were used as a secondary antibody.

## Statistical analysis

Graphs were generated using GraphPad software, and all data were represented by mean ± SEM. T-tests were used for statistical analysis when there were only two groups, and one-way ANOVA was used when there were three or more groups. The number of all biological replicates in the experiment was three (n=3). Significance was indicated using "*" $p < 0.05$, "**" $p < 0.01$, "***" $p < 0.001$, "****" $p < 0.0001$.

# Results

## Cloning and expression analysis of the goat LIPG gene

As the lack of validation sequence of goat *LIPG* gene on NCBI database, we successfully cloned the LIPG gene mRNA sequence for the first time in this study. The predicted sequence of *LIPG* (GENE BANK: XM_005697191.3) was used to design primers (S1 Table). The full length sequence of the cloned *LIPG* gene is 2,131 bp, including 1,503 bp CDs, 94 bp 5′ untranslated region (5′UTR), and 534 bp 3′UTR, encoding 500 amino acids (Fig 1A). The phylogenetic tree of LIPG amino acids constructed by MEGA5.05 showed that goat LIPG was closely related to sheep, but distant from human and mouse (Fig 1B). Expression analysis revealed that LIPG mRNA is most abundantly expressed in the small intestine, followed by heart, liver and kidney (Fig 1C). Additionally, LIPG expression varied throughout the differentiation process of goat preadipocytes, reaching its lowest level on day 6 and peaking on day 8 on 8th day (Fig 1D).

## Knockdown of LIPG inhibits lipid deposition in goat intramuscular preadipocytes

To knock down the expression of LIPG, two siRNAs were synthesized, achieving knockdown efficiencies of 59% and 42%, respectively (Fig 2A). In addition, siRNA-1 effectively knocked down LIPG expression at the protein level (Fig 2B–2C). SiRNA-1 was used in the following experiment. At the morphology, compared with the negative control group (NC), the LIPG-silenced group (Si-LIPG) exhibited fewer lipid droplets, as labeled by oil red O stained. (Fig 2D-2E) and BODIPY staining (Fig 2F–2G). Although no significant difference was achieved, LIPG knocking down also reduced the intracellular TAG content (Fig 2H). The increase in intramuscular fat mass may be the result of the generation of new adipocytes and/or an increase in triglyceride deposition by individual cells [31]. Previous studies have shown that

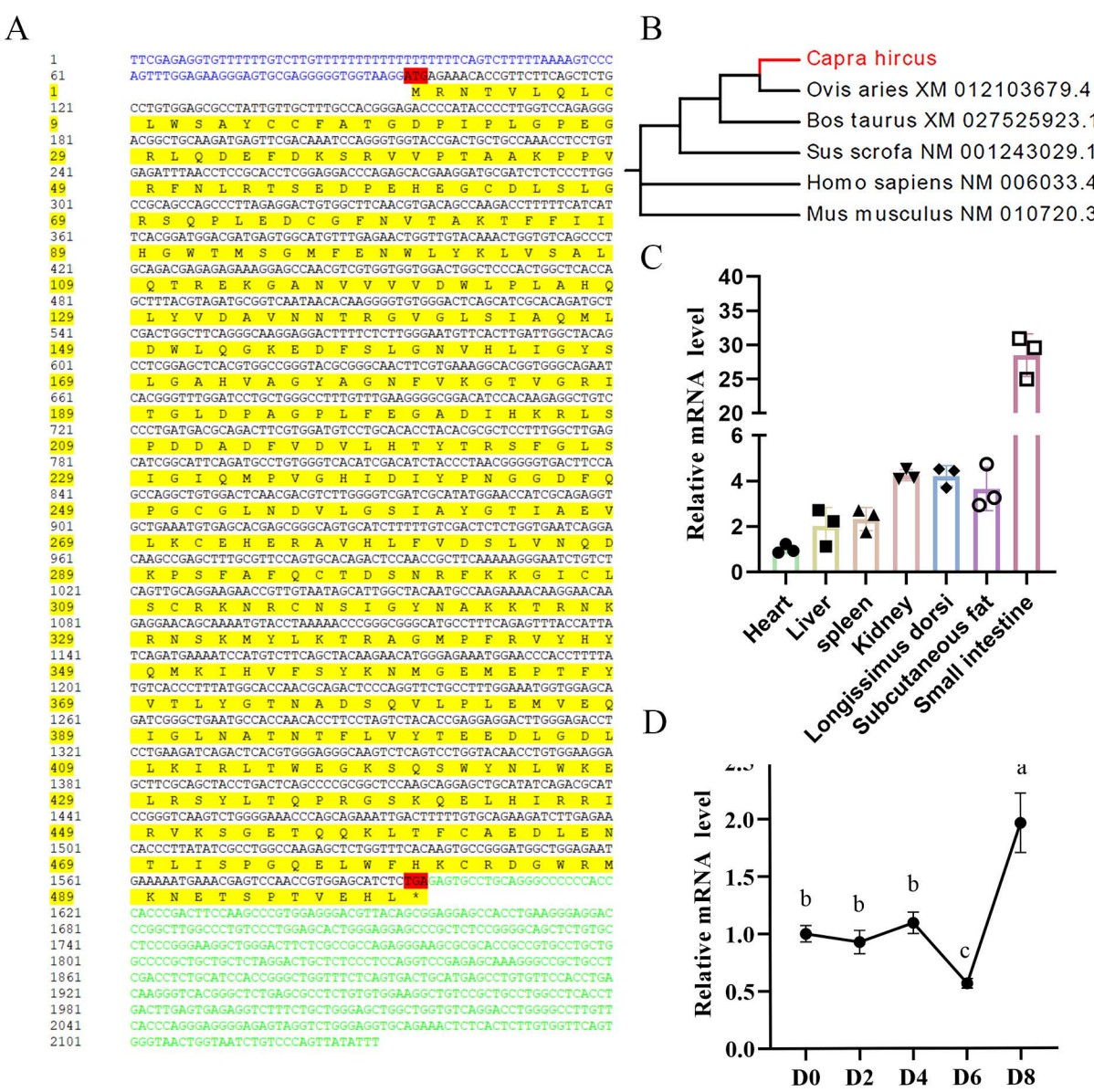

**Fig 1. Cloning, genetic relationship analysis, tissue expression profile and time sequence expression profile of goat *LIPG* gene.** (A) The nucleotide sequence and the translated amino acid sequence of goat *LIPG* (blue indicates 5'UTR; green indicates 3'UTR; black indicates ORF; yellow indicates amino acid sequence; red indicates initiation codon and stop codon; * Represents the stop codon). (B) LIPG amino acid sequence phylogenetic tree was constructed by MEGA5.05. (C) The *LIPG* mRNA level in heart, liver, spleen, kidney, longissimus dorsi, subcutaneous fat and small intestine, **n** = 3. Ubiquitously expressed prefoldin like chaperone (*UXT*) as the internal reference gene; (D) The *LIPG* mRNA level at day 0, 2, 6 and 8 in induced differentiation intramuscular adipocyte (n = 3).

LIPG supports cell proliferation in triple negative breast cancer cells [11]. Therefore, we speculate that functional loss of *LIPG* may change the proliferative capacity of goat intramuscular preadipocytes. In the process of inducing the intramuscular preadipocytes to differentiate into adipocytes, the proliferation capacity of the cells was decreased by EDU staining (Fig 2I–2J).

Concurrently, knockdown of the *LIPG* in intramuscular preadipocytes significantly increased mRNA expression of adipose triglyceride lipase (*ATGL*), acyl-CoA oxidase 1

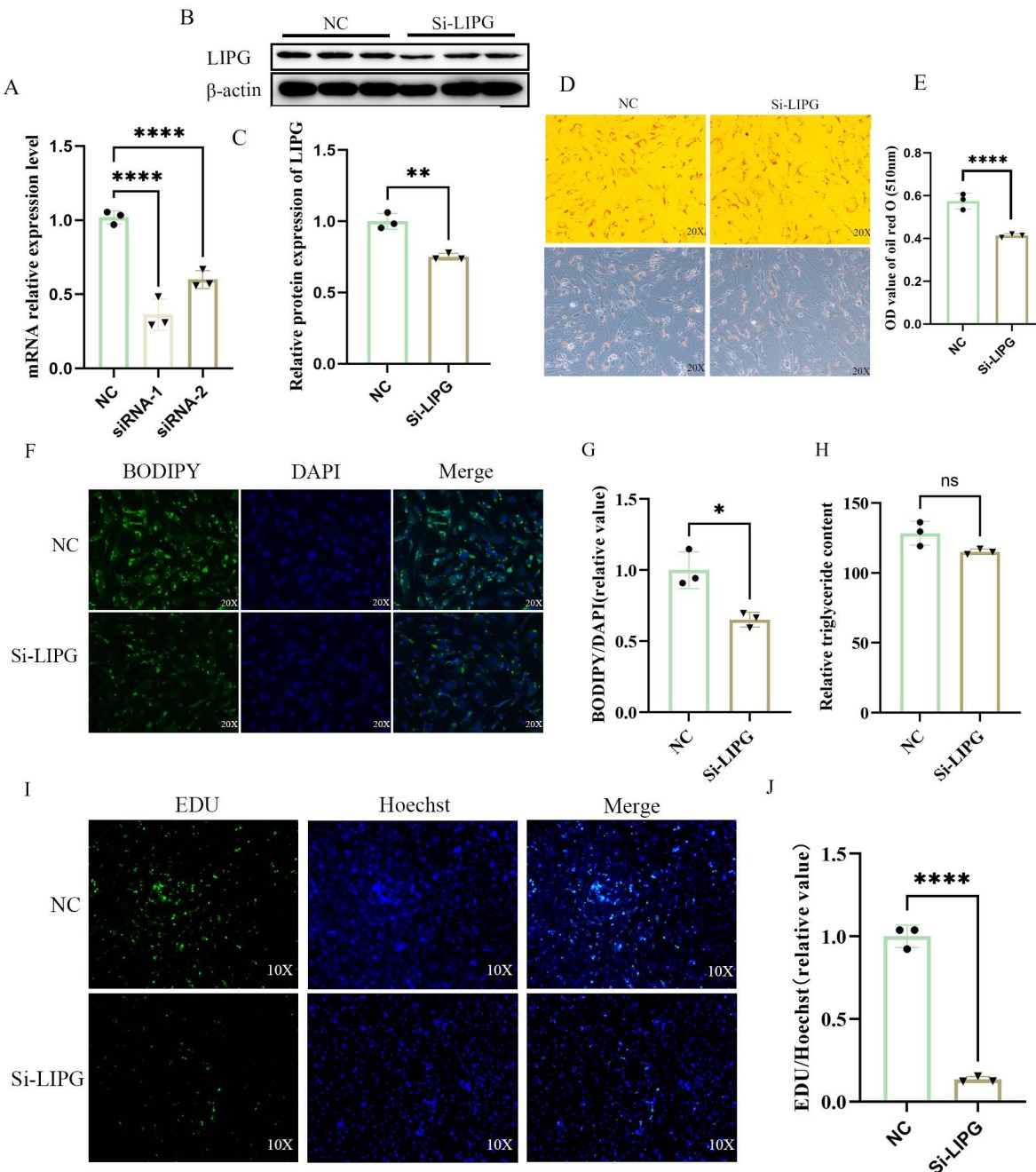

**Fig 2. Knockdown of *LIPG* inhibits lipid deposition in goat intramuscular preadipocytes.** (A) The knockdown efficiency detection of *LIPG* by real-time quantitative PCR (RT-qPCR). (B) LIPG protein expression levels were detected in the NC and Si-LIPG groups using WB. (C) The quantification analysis of WB using Image-Pro Plus 6.0. (D) The Oil Red O staining between negative control groups (NC) and siRNA treatment groups (Si-LIPG) in intramuscular preadipocyte. (E) The quantitative analysis of Oil Red O staining was indicated by the OD value at 510 nm. (F) The BODIPY staining between NC groups and Si-LIPG groups. Lipid droplets were visualized with BODIPY 493/503 staining (green). Nuclei were stained with DAPI (blue). (G). The fluorescence intensity was measured by Image-Pro Plus 6.0. (H) Intracellular triglycerides (TAG) levels were quantified with the triglyceride quantification assay kit. (I) EDU staining between NC groups and Si-LIPG groups; Green fluorescence (EDU) represents an EDU positive cell and blue fluorescence (Hoechest) represents the nucleus. (J) Quantitative analysis of the number of EDU positive cells by Image-Pro Plus 6.0. Data are shown as mean ± SEM, n = 3, * P < 0.05, ** P < 0.01, *** P < 0.001, **** P < 0,0001.

(*ACOX1*), fatty acid desaturase 1 (*FADS1*), long-chain fatty acid family member 6 (*ELOVL6*), while significantly decreasing the expression levels of fatty acid synthase (*FASN*), lipoprotein lipase (*LPL*), hormone-sensitive lipase (*HSL*), carnitine palmitoyltransferase 1A (*CPT1A*), carnitine palmitoyltransferase 1B (*CPT1B*), and fatty acid binding protein 3 (*FABP3*) (Fig 3). However, the expression of glycerol-3-phosphate acyltransferase 4 (*AGPAT6*), diacylglycerol O-acyltransferase 1 (*DGAT1*), acyl-CoA synthetase long chain family member 1 (*ACSL1*), acyl-CoA synthetase short chain family member 2 (*ACSS2*), acetyl-CoA carboxylase (*ACC*), and *CD36* were not obviously changed by knockdown of LIPG (Fig 3). These results suggest that *LIPG* gene deletion significantly decreased IMF lipid deposition by reducing intramuscular preadipocyte proliferation and lowering lipid content.

## Overexpression of LIPG promotes lipid deposition in goat intramuscular preadipocytes

The OE-LIPG vector significantly upregulated both mRNA and protein expression levels of LIPG (Fig 4A-4C). Compared with the control groups (NC), overexpression of *LIPG* groups (OE-LIPG) significantly increased cellular adipogenesis by Oil Red O staining (Fig 4D–4E) and BODIPY staining (Fig 4F-4G). The intracellular TAG content was elevated, although not statistically significant (Fig 4H). Furthermore, functional acquisition of *LIPG* significantly increased the ability of goat intramuscular preadipocytes to proliferate (Fig 4I–4J). *LIPG* overexpression did not affect the expression levels of *AGPAT6*, *DGAT1*, *ACSL1*, *ACSS2*, *ACC*, *CD36* (Fig. 5). However, it significantly increased the expression of *LPL*, *HSL*, *CPT1A*, *CPT1B*,

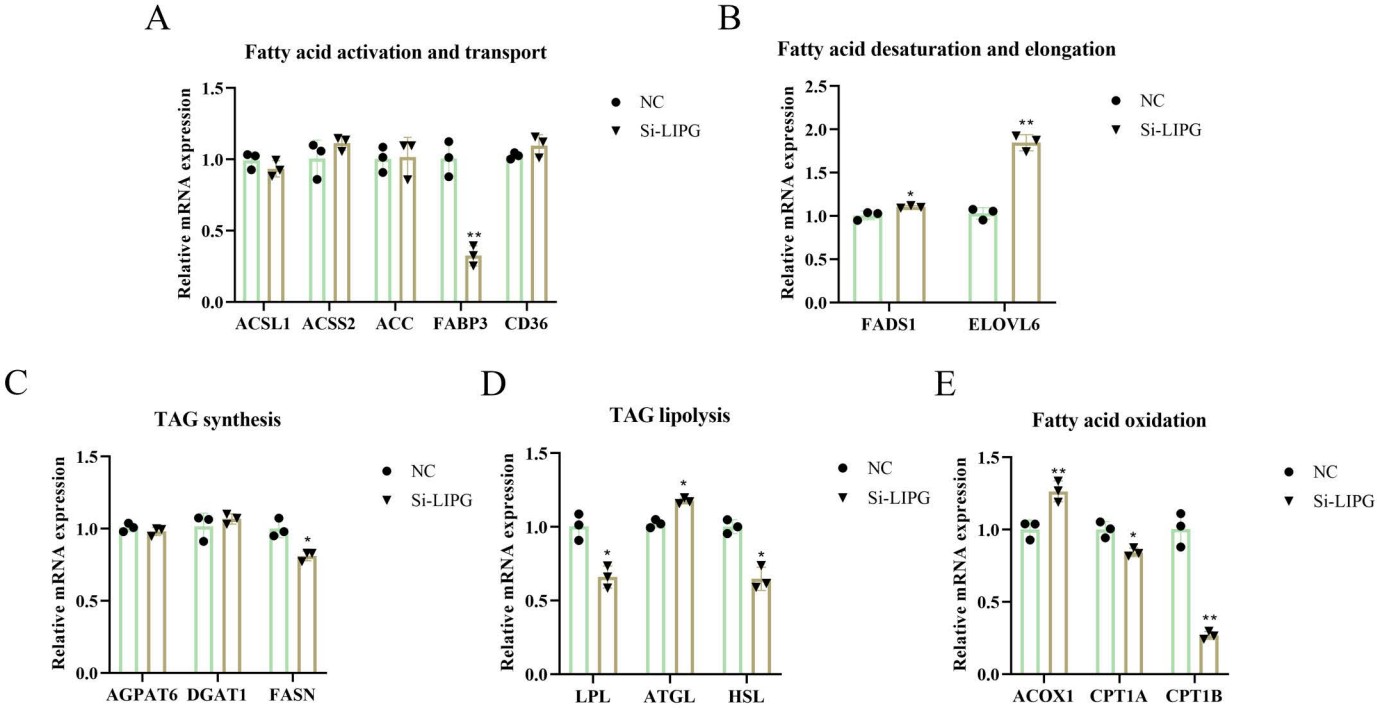

**Fig 3. Knockdown of *LIPG* altered the expression level of genes related to lipid metabolism.** (A) Fatty acid activation and elongation related genes. (B) Fatty acid desaturation and elongation related genes. (C) Triglycerides (TAG) synthesis related genes. (D) TAG lipolysis related genes. (E) Fatty acid oxidation related genes. Data are shown as mean ± SEM, * P < 0.05, ** P < 0.01, *** P < 0.001.

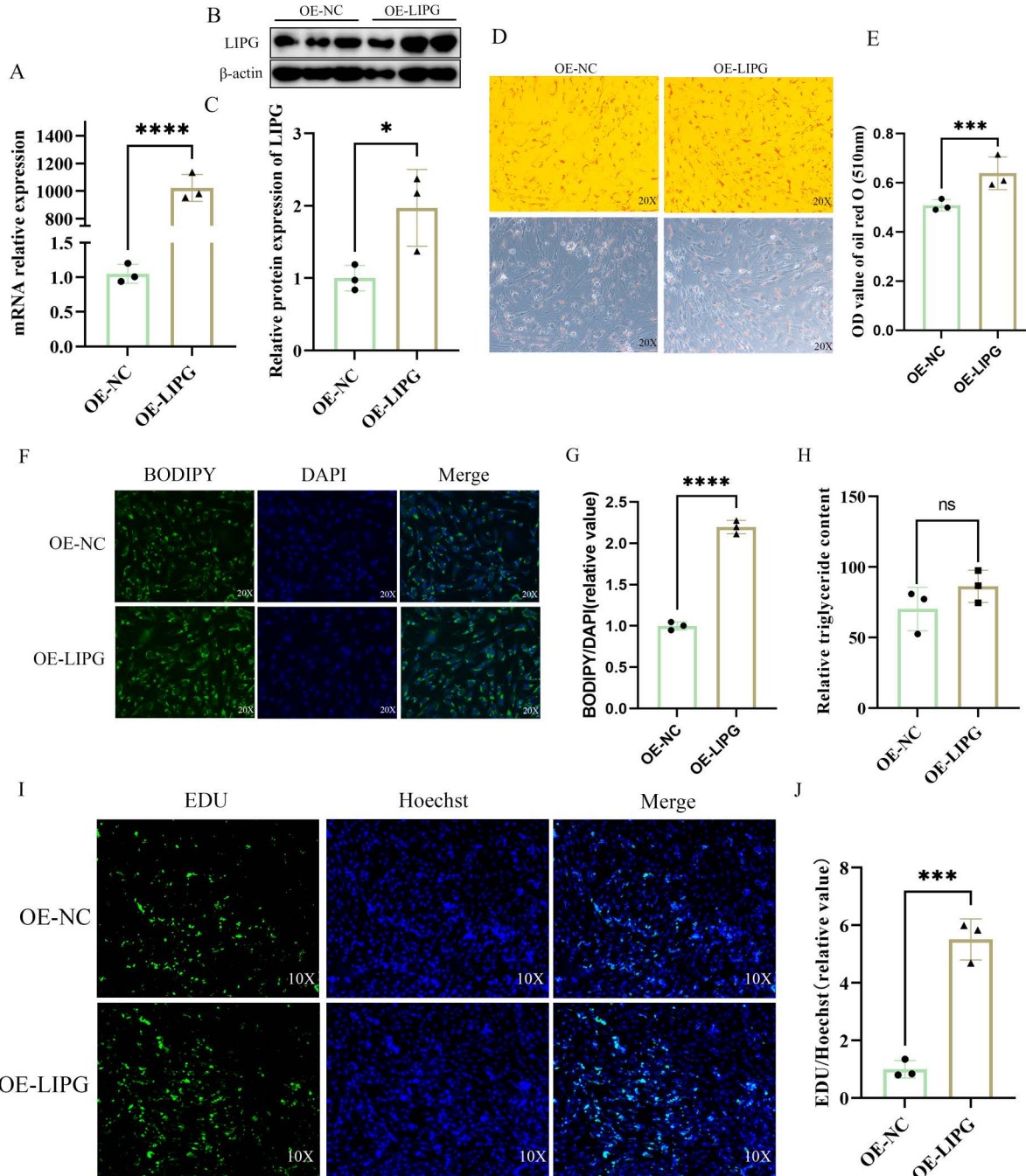

**Fig 4. Overexpression of *LIPG* promoted lipid deposition in goat intramuscular preadipocyte deposition.** (A) The overexpression efficiency of *LIPG* by RT-qPCR. (B) LIPG protein expression levels were detected in the OE-NC and OE-LIPG groups by WB. (C) The quantification analysis of WB using Image-Pro Plus 6.0. (D) The Oil Red O staining between OE-NC groups and OE-LIPG groups in intramuscular preadipocyte cells. (E) The quantitative analysis of Oil Red O staining was indicated by the OD value at 510 nm. (F) The BODIPY staining between OE-NC groups and OE-LIPG groups. Lipid droplets were visualized with BODIPY 493/503 staining (green). Nuclei were stained with DAPI (blue). (G). The fluorescence intensity was measured by Image-Pro Plus 6.0. (H) Intracellular TAG levels were quantified with the triglyceride quantification assay kit. (I) EDU staining between NC groups and OE-LIPG groups; Green fluorescence (EDU) represents an EDU positive cell and blue fluorescence (Hoechst) represents the nucleus. (J) Quantitative analysis of the number of EDU positive cells by Image-Pro Plus 6.0. Data are shown as mean ± SEM, n = 3, * P < 0.05, ** P < 0.01, *** P < 0.001, ****P < 0.0001.

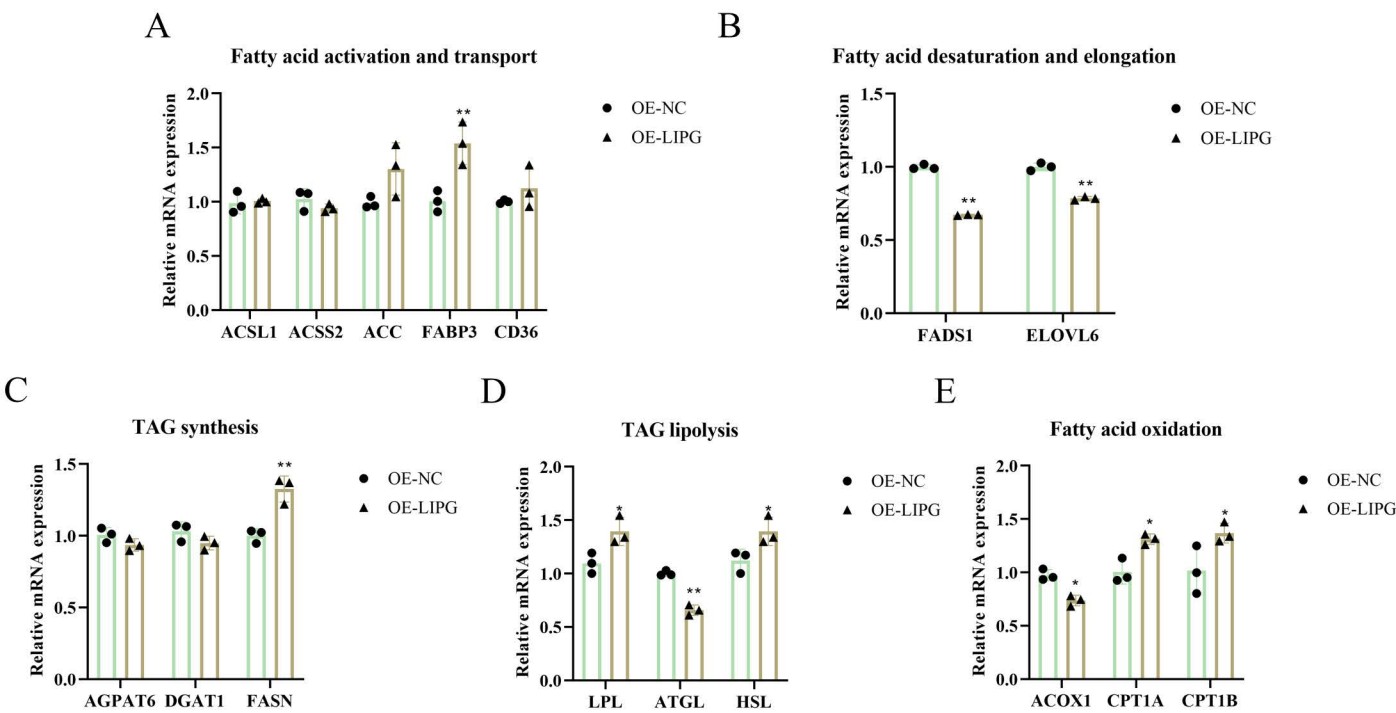

**Fig 5. Overexpression of *LIPG* altered the expression level of genes related to lipid metabolism.** (A) Fatty acid activation and elongation related genes. (B) Fatty acid desaturation and elongation related genes. (C) TAG synthesis related genes. (D) TAG lipolysis related genes. (E) Fatty acid oxidation related genes. Data are shown as mean ± SEM, * P < 0.05, ** P < 0.01, *** P < 0.001.

*FABP3*. Conversely, the expression levels of *ATGL*, *ACOX1*, *FADS1* and *ELOVL6* were significantly reduced upon LIPG overexpression (Fig 5). These results show that overexpression of LIPG promotes lipid deposition in goat intramuscular preadipocytes. To investigate the role of LIPG activity in promoting lipid deposition induced by overexpression of LIPG, we added orlistat to intramuscular preadipocytes overexpressing the LIPG gene. Addition of orlistat did not attenuate the increase in lipid deposition caused by overexpression of LIPG (S1 Fig), suggesting that LIPG-promoted lipid deposition does not act through LIPG activity.

## Knockdown of LIPG affects the mRNA transcriptional profile of goat intramuscular preadipocytes

To further clarify the molecular mechanism by which LIPG regulates lipid deposition in intramuscular preadipocytes, we constructed a transcription profile of *LIPG* silencing by RNA-seq. Principal Components Analysis (PCA) analysis showed that the transcriptome data of the negative control groups (NC) and the *LIPG* knockdown groups (Si-LIPG) were divided into two independent clusters (n = 3) (S2 Fig). The reliability of the sequencing data was verified by RT-qPCR (S3 Fig). A total of 1695 differentially expressed genes (DEGs) were identified between NC groups and Si-LIPG groups (p<0.05, Fold change>1.5), with 1064 genes significantly upregulated and 631 genes differentially downregulated (Fig 6A). The most significantly upregulated was the G Protein Pathway Suppressor 1 (GPS1) gene, followed by the Coiled-Coil Domain Containing 85B (Delta-Interacting Protein A) CCDC85B gene. The Top 50 genes with the most significant differences are listed in supplemental Table 2 (S2

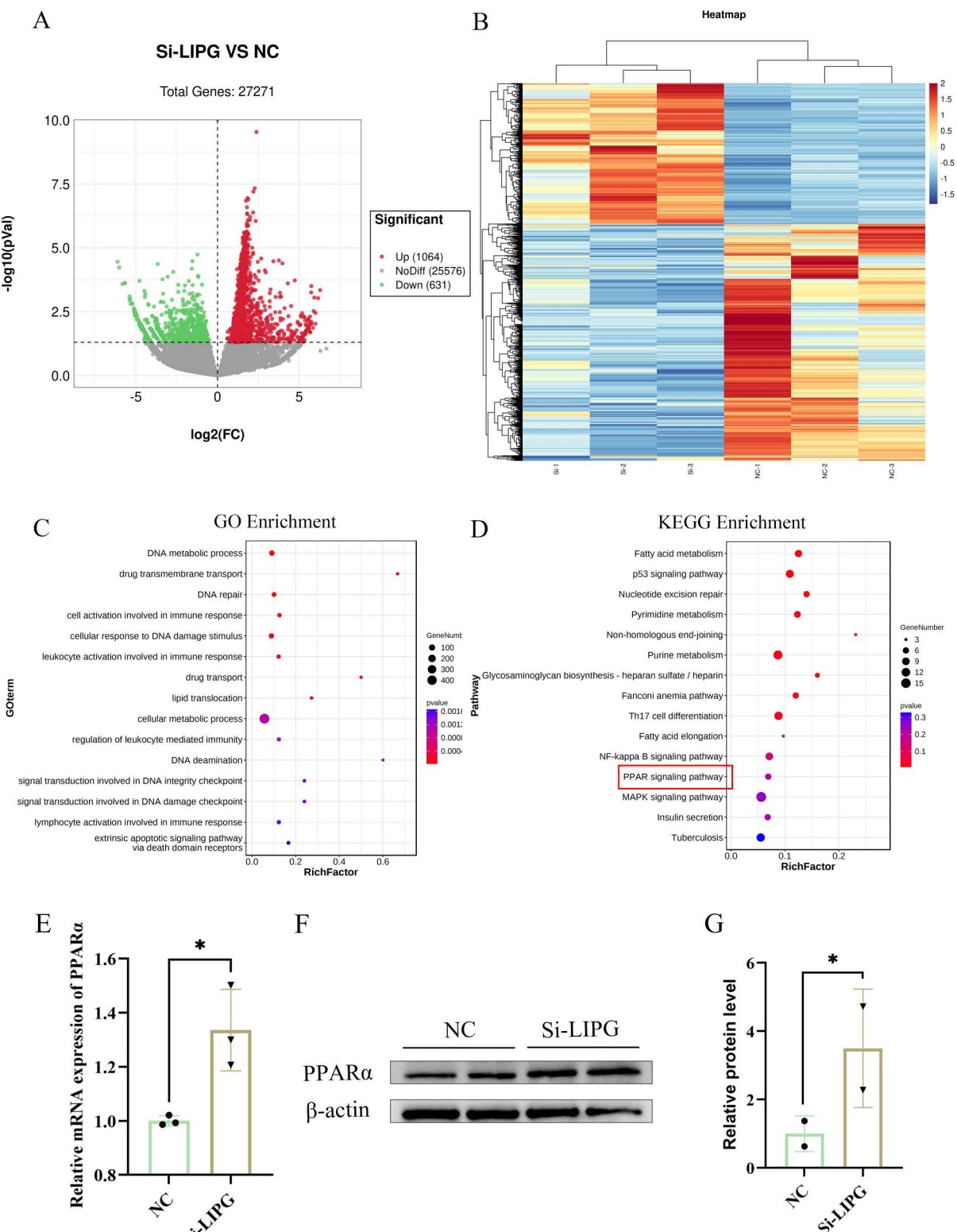

**Fig 6. Knockdown of *LIPG* affected the mRNA transcript profiles.** (A) RNA-seq volcano plot of significantly differential expression genes (DEGs) in Si-LIPG (n = 3) vs NC (n = 3) goat intramuscular preadipocytes, Red and green dots denote upregulated and down-regulated genes, respectively. (B) The heat map shows the relative levels DEGs. (C) The GO pathway analysis of related DEGs. (D) The KEGG pathway

analysis of related DEGs. (D) The PPARα mRNA expression of silencing LIPG. (E) Western blot detection of protein levels in NC groups (NC) and LIPG interference groups (Si-LIPG) (n = 2). (F) Quantification of Western blot results by Image-Pro Plus 6.0.

Table). To visualize the whole effect of the DEGs expression, the hierarchical cluster analysis in the form of a heat map is shown below (Fig 6B). These DEGs were significantly enriched in GO terms such as lipid translocation, cellular metabolic process, heparin metabolic process, etc. (Fig 6C). And there were significantly enriched in signaling pathways such as fatty acid metabolism, p53 signaling pathway, fatty acid elongation, NF-kappa B signaling pathway, PPAR signaling pathway, MAPK signaling pathway (Fig 6D) by KEGG enrich analysis. PPARα signaling pathway is closely related to energy metabolic processes such as fatty acid oxidation and triglycerides [33]. The silencing of *LIPG* changed the PPARα signaling pathway by KEGG enrich analysis, and significantly increased the expression of PPARα by RT-qPCR and Western Blot (Fig 6D-6 F). In addition, Acyl-CoA synthetase long chain family member 5 (*ACSL5*), perilipin 4 (*PLIN4*), FABP3, sterol carrier protein 2 (*SCP2*) and phospholipid transfer protein (*PLTP*) were enriched to PPAR signaling pathway.

### PPARα signaling pathway participates in LIPG regulation of lipid deposition in goat intramuscular preadipocytes

To verify the role of PPARα signaling pathway in mediating the function of *LIPG* knockdown, we added the PPARα agonist WY-14643 (50 μM) [34] to the intramuscular preadipocytes while overexpression of *LIPG*. The results of BODIPY staining showed that the OE-LIPG+WY14643 groups had more lipid droplets than the WY-14643 groups and less than the OE-LIPG groups (Fig 7A-7C). Meanwhile, the activation of PPARα reversed the reversed the altered expressions of *DGAT1*, *DGAT2*, *LPL* and *ACOX1* induced by *LIPG* overexpression (Fig 7C). Taken together, these data reveal that increased lipid deposition due to overexpression of *LIPG* can be reversed by PPARα agonist.

### Discussion

The involvement of LIPG in human diseases and lipid metabolism has been well studied [19,26,35,36]. Overexpression of *LIPG* increases intracellular triglyceride content in HAEC cells and promotes proliferation of triple-negative breast cancer cells [11,16,37]. However, the specific function of LIPG in regulating lipogenesis and lipid deposition in goat IMF remains unclear. This study is novel that we screened and validated that LIPG regulates the proliferation and lipid deposition of goat intramuscular preadipocytes, at least partly, through the control of PPARα signaling pathway (Fig 8). These data may improve our understanding about the mechanism underlying IMF formation in goat.

Endothelial Lipase (LIPG) was named for its initial identification in Human Umbilical Vein Endothelial Cells (HUVEC) and was subsequently found to be expressed in highly metabolic tissues such as heart and liver [38,39]. Our study confirmed that *LIPG* is expressed in a variety of tissues in goats, but interestingly its expression levels was highest in the small intestine compared to the other tissues tested. Considering the HDL is synthesized in the small intestine in addition to the liver [40], the highly expression of LIPG may be responded to the metabolism of small intestine-derived HDL.

In the previous study, LIPG downregulation decreased lipid synthesis capacity and proliferation of breast cancer (BCa) cells [16]. Our findings are consistent with this, as knocking down LIPG inhibited lipid deposition and cell proliferation in goat intramuscular preadipocytes.

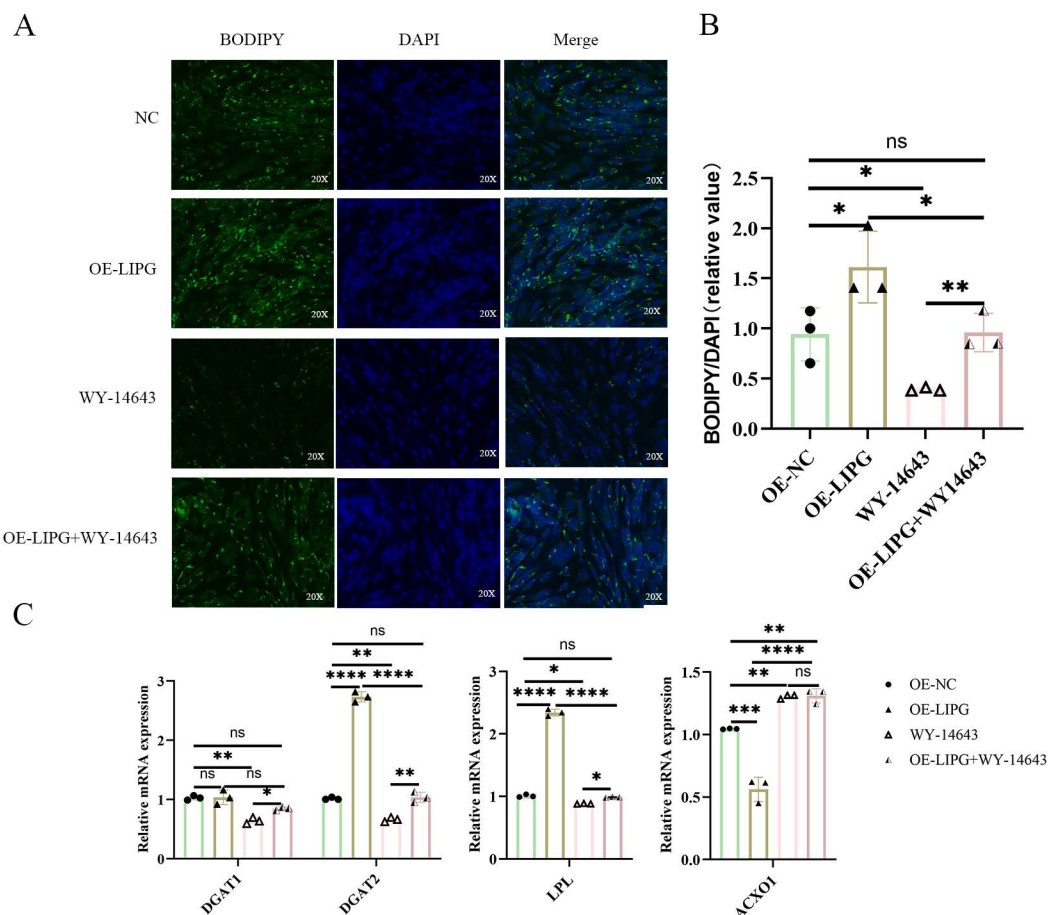

**Fig 7. LIPG-induced lipid deposition is partially dependent on the PPAR α signaling pathway in goat intramuscular preadipocytes.** (A) BODIPY 493/503 staining (green). Nuclei were stained with DAPI (blue). (B) BODIPY staining were measured by Image-Pro Plus 6.0. (C) Relative mRNA expression of lipid metabolism genes. Data are shown as mean ± SEM, n = 3, * < 0.05, ** < 0.01, *** < 0.001, ****<0.0001.

Knockdown LIPG reduced de novo fatty acid synthesis by down-regulating the expression of FASN, a key enzyme in de novo fatty acid synthesis [41]. It also reduced lipid droplet content by increasing ATGL to lipolysis ATG into diacylglycerol (DAG) and FAs [42,43]. However, HSL, the main decomposing agent of DAG [42], was not reduced. It is possible that the lipolysis product FAs is involved in the regulation of HSL as a signaling molecule. We are puzzled that overexpression of LIPG increased LPL expression in goat intramuscular preadipocytes. Previous studies have shown that Lipg$^{-/-}$ mice have higher levels of LPL expression than WT mice, and it has been demonstrated that LIPG and LPL collaborate to promote efficient triglyceride (TG)-rich lipoproteins (TRLs) lipolysis [36]. The differences between our results and the reported in mice may be due to cell-individual differences and species specific differences between mice and goats. Obtaining knock-out goats for in vivo studies of LIPG gene function related to IMF deposition is likely not feasible due to the high cost and concern for animal welfare. Taken together, the promotion of cellular lipid deposition by LIPG observed in goat intramuscular preadipocyte may be achieved by increasing de novo fatty acid synthesis and decreasing lipolysis.

In this study, the PPAR signaling pathway was predicted to mediate the effects of *LIPG* silencing in goat intramuscular preadipocytes based on KEGG pathway analysis. Knockdown

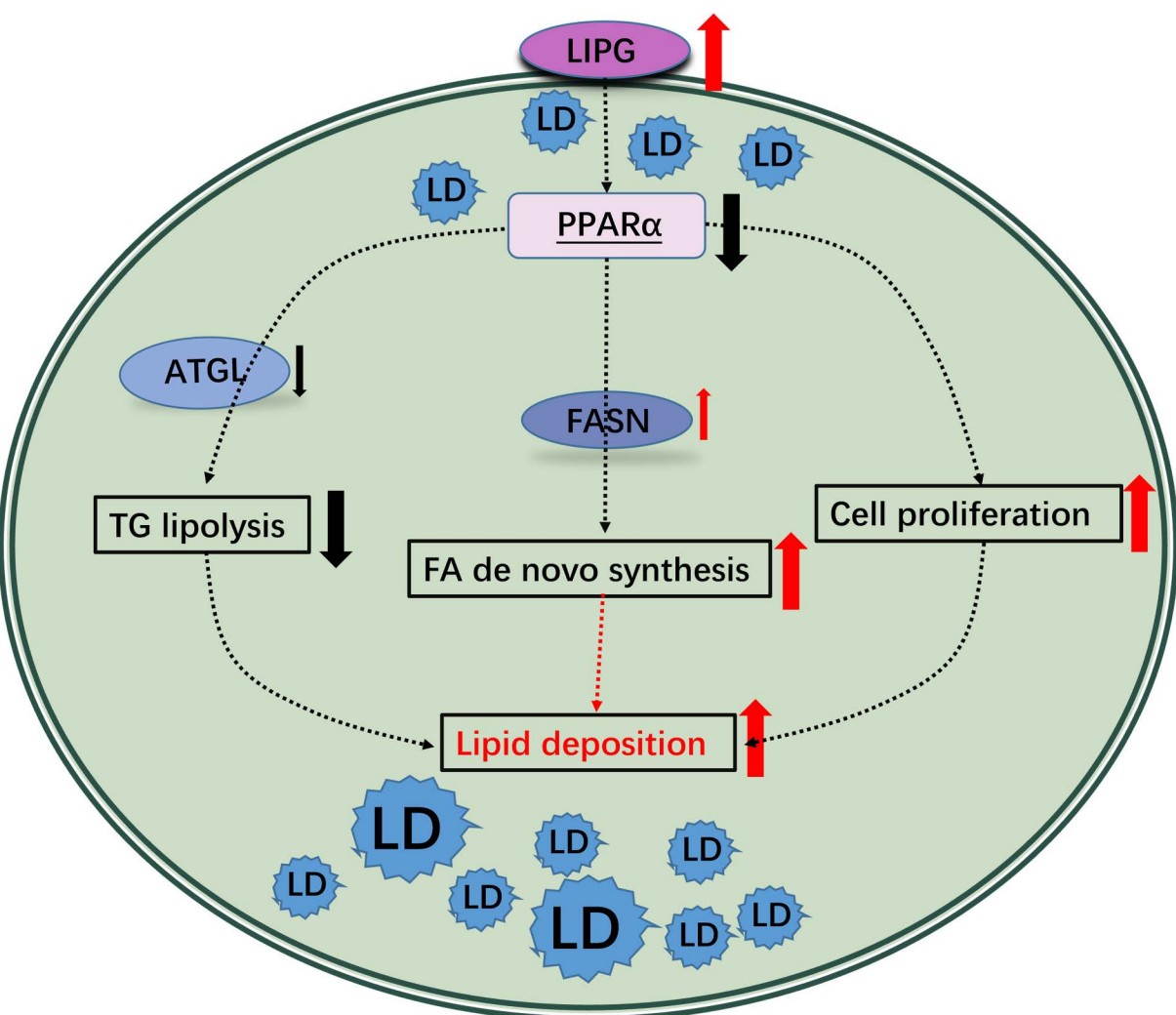

**Fig 8. LIPG regulates goat intramuscular preadipocyte lipid deposition through the PPAR α signaling pathway.** LIPG regulates the PPARα signaling pathway, which further regulates FA de novo synthesis, TAG lipolysis, cell proliferation. This regulation may achieve through the modulation of target genes such as *FASN*, *ATGL*.

and overexpression of *LIPG* elicited significant changes of *FABP3*, *ACXO1*, *FASN*, *LPL*, *ATGL*, and *HSL*, which have been previously reported to be regulated by the PPARα signaling pathway [44] mostly by promoter targeting [44–46]. Rescue experiments were conducted by adding PPARα agonists WY-14643 to cells overexpressing *LIPG*. Notably, the results exhibited partial restoration of the phenotype and gene expression alterations caused by *LIPG* overexpression when treated with the PPARα agonist WY-14643. This finding supported our earlier demonstration that LIPG exerts regulatory functions through the PPAR signaling pathway, which aligns well with previous study that LIPG activates PPARα in Bovine aortic endothelial cells (BAEC) [47]. CPT1 may be regulated by PPARa, AMPK, MAPK and other signaling pathways [48–50]. Knockdown of LIPG increased the expression level of PPARa but decreased the expression level of CPT1A and CPT1B, the target genes of PPARa, suggesting that its LIPG regulates lipid deposition in goat intramuscular preadipocytes partly through the PPARa signaling pathway.

Despite these insights, we recognize the limitations of this study, particularly the lack of direct evidence on how LIPG regulates the PPAR signaling pathway. It is well known that LIPG hydrolyzes HDL to release FAs, which can be taken up and utilized by neighboring tissues and cells [12,15,51]. Furthermore, FAs can activate the PPARα signaling pathway as ligands [52]. We hypothesized that LIPG release FAs through lipolysis of lipid-rich proteins to provide the ligands for PPARα, regulates cellular fatty acid synthesis, triglyceride lipolysis, fatty acid oxidation, cell proliferation and other physiological activities. However, our data from the orlistat experiment suggest that lipid accumulation changes associated with LIPG overexpression may not be solely dependent on its enzymatic activity, as orlistat treatment did not reverse the increased lipid accumulation. This raises the possibility that LIPG could also regulate lipid deposition through non-enzymatic mechanisms, particularly in goat intramuscular preadipocytes. While these findings contribute to our understanding, further experimental validation is needed to clarify the precise mechanisms—both enzymatic and potential non-enzymatic—through which LIPG regulates lipid metabolism and the PPARα pathway.

## Conclusions

In conclusion, the coding sequence of goat LIPG spans 1,503 base pairs and exhibits the high expression levels in the heart, liver, and kidney tissues. Our findings suggest that LIPG play a positive regulatory role in intramuscular fat (IMF) content by promoting both lipid deposition and preadipocyte proliferation. Furthermore, our analysis indicates the involvement of the PPARα signaling pathway in mediating the effects of LIPG on cellular lipid accumulation in goat intramuscular preadipocytes. These data will provide a theoretical basis for elucidating the regulatory mechanism of goat IMF.

## Supporting information

**S1 Table. Summary of genes, peimers and product sizes for RT-qPCR.**
(PDF)

**S2 Table. The top-50 gene of DEGs.**
(PDF)

**S1 Fig. Effect of adding orlistat on overexpression of LIPG.**
(TIF)

**S2 Fig. Principal component analysis (PCA) groups Si-LIPG and NC.**
(TIF)

**S3 Fig. Transcriptome sequencing results validated by RT-qPCR).**
(TIF)

**S1 Raw Images. Raw images of Western Blot experiments for this study.**
(PDF)

## Author contributions

**Conceptualization:** Lian Huang, JiangJiang Zhu Yinmei Tang, Yaqiu Lin, Hua Xiang.

**Data curation:** Yinggui Wang, JiangJiang Zhu, Yaqiu Lin, Lian Huang.

**Formal analysis:** Lian Huang, JiangJiang Zhu.

**Funding acquisition:** JiangJiang Zhu, Yong Wang.

**Investigation:** JiangJiang Zhu, Changheng Yang, Yong Wang.

**Methodology:** Yinggui Wang, Lian Huang, JiangJiang Zhu, Changheng Yang, Hua Xiang.

**Project administration:** Lian Huang, JiangJiang Zhu, Yong Wang.

**Resources:** Yinggui Wang, Lian Huang, JiangJiang Zhu, Yong Wang.

**Software:** Yinggui Wang, Wenyang Zhang.

**Supervision:** Lian Huang, JiangJiang Zhu, Yaqiu Lin, Yong Wang.

**Validation:** Yinggui Wang, Yinmei Tang.

**Visualization:** Yinggui Wang, Lian Huang, Wenyang Zhang, Yinmei Tang.

**Writing – original draft:** Yinggui Wang.

**Writing – review & editing:** Yinggui Wang, Lian Huang, JiangJiang Zhu, Hua Xiang.

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
