## [Decision Letter · Decision Letter 0]

28 Nov 2023

PONE-D-23-34403LIPG promotes lipid deposition through PPARα signaling pathway in goat intramuscular preadipocytePLOS ONE

Dear Dr. Zhu,

Thank you for submitting your manuscript to PLOS ONE. After careful consideration, we feel that it has merit but does not fully meet PLOS ONE’s publication criteria as it currently stands. Therefore, we invite you to submit a revised version of the manuscript that addresses the points raised during the review process.

We look forward to receiving your revised manuscript.

Kind regards,

Jérôme Robert, PhD

Academic Editor

PLOS ONE

- https://www.mdpi.com/1422-0067/24/17/13415

In your revision ensure you cite all your sources (including your own works), and quote or rephrase any duplicated text outside the methods section. Further consideration is dependent on these concerns being addressed.

 [Jiangjiang Zhu was supported by Sichuan Science and Technology Program (2021YFYZ0003)；Jiangjiang Zhu was supported by Zhejiang Science Technology Program (2022C04017)；Hua Xiang was supported by Sichuan Science and Technology Program (2022NSFC0082)；Yinggui Wang was supported by Graduate innovative scientific research project of Southwest Minzu University (ZD2022726)；].  

5. PLOS requires an ORCID iD for the corresponding author in Editorial Manager on papers submitted after December 6th, 2016. Please ensure that you have an ORCID iD and that it is validated in Editorial Manager. To do this, go to ‘Update my Information’ (in the upper left-hand corner of the main menu), and click on the Fetch/Validate link next to the ORCID field. This will take you to the ORCID site and allow you to create a new iD or authenticate a pre-existing iD in Editorial Manager. Please see the following video for instructions on linking an ORCID iD to your Editorial Manager account: https://www.youtube.com/watch?v=_xcclfuvtxQ".

6. We note that Figure(s) 2B, 2D, 2G, 4B, 4D, 4G, 6B and 7D in your submission contain copyrighted images. All PLOS content is published under the Creative Commons Attribution License (CC BY 4.0), which means that the manuscript, images, and Supporting Information files will be freely available online, and any third party is permitted to access, download, copy, distribute, and use these materials in any way, even commercially, with proper attribution. For more information, see our copyright guidelines: http://journals.plos.org/plosone/s/licenses-and-copyright.

a. You may seek permission from the original copyright holder of Figure(s) 2B, 2D, 2G, 4B, 4D, 4G, 6B and 7D to publish the content specifically under the CC BY 4.0 license. 

Reviewers' comments:

Reviewer's Responses to Questions

**Comments to the Author**

1. Is the manuscript technically sound, and do the data support the conclusions?

Reviewer #1: Partly

Reviewer #2: Partly

Reviewer #3: No

2. Has the statistical analysis been performed appropriately and rigorously? 

Reviewer #1: Yes

Reviewer #2: I Don't Know

Reviewer #3: I Don't Know

3. Have the authors made all data underlying the findings in their manuscript fully available?

Reviewer #1: Yes

Reviewer #2: Yes

Reviewer #3: Yes

4. Is the manuscript presented in an intelligible fashion and written in standard English?

Reviewer #1: No

Reviewer #2: No

Reviewer #3: No

5. Review Comments to the Author

Reviewer #1: The authors cloned goat LIPG and examine the impact of LIPG overexpression and silencing on lipid accumulation, expression of lipid metabolism related genes and cell proliferation in goat intramuscular preadipocytes. Additionally, the authors examine by RNAseq the impact of LIPG-knock down on the transcriptome of the preadipocytes.

The results show that EL (partially via PPARa) modulates expression of lipogenic genes, lipid accumulation and proliferation of goat intramuscular preadipocytes.

The manuscript is very poorly written with very poor English with a lot of grammar and syntax errors which make the reading very difficult.

Experimental details regarding Oil Red, BODIPY, and EDU staining are missing: the authors should provide information on the time frame between siRNA- and plasmid- transfection and staining with Oil Red, BODIPY, and EDU; were the cells incubated in a full medium with 10% FCS?

Page 6 lane 115; cells were treated with triglyceride lysate? What does this mean?

LIPG Western blot (WB) of untreated cells as well as of LIPG silenced and overexpressing cells should be performed and shown; WB of cell lysates and of heparin medium should be shown.

It is not stated which statistical tests were used for the analyses.

The authors should discuss why LPL is upregulated after LIPG overexpression. Previous studies have shown upregulation of EL in WAT of LPL-KO mice.

Description of Fig. 7 is confusing and something is not correct. The authors show the impact of LIPG silencing on PPARa, then the impact of overexpression on lipid accumulation (BODIPY) and also graph with EDU results, but state in the legend it is BODIPY. Fig. 7F shows upregulated ATGL after LIPG overexpression which is contradictory to Figs 3 and 5.

Page 13, lane 305: It is not correct that LIPG was identified first in endothelial-type adipocytes; it was in HUVEC.

Page 14, lane 331: The reference 48 is wrong; this paper is on HL and not on LIPG and PPARa.

Reviewer #2: The manuscript of Zhu et al describes the effects of LPIG on cultured goat intramuscular (pre)adipocytes and they suggest that LIPG enhances lipid deposition in these cells via effects on PPARa signaling. I do have several comments on this work.

1. What was the rationale to study the role of LIPG in goat specifically? Is it indeed as mentioned on page 3 to improve goat meat quality? I bet there will be better targets than LIPG and alternative methods to do this. Please explain in more detail.

2. How do the results in the current paper compare to data available from, for instance, LIPG knockout mice? Do they show similar effects on adipose tissue? This should be discussed in the discussion.

3. I’m completely lost in the way the authors culture the cells and performed their experiments. Firstly, how are they sure that they collected the pre-adipocytes from the big piece of muscle they collected from goats? I’d advise to show some typical pre-adipocyte markers before and directly after isolation form the muscle as well as after the differentiation. Secondly, is adding oleic acid enough to get adipocytes or are the cultured cells in the end still pre-adipocytes? Looking at the figures I get the impression that these cells are still the latter. So why use this protocol and not more established ones like adding a PPARgamma agonist and insulin? Can the authors also provide data on whether the differentiation worked by for instance showing bona fide adipogenesis as well as proliferation markers at various time-points? Thirdly, the information on when the exact experiments were performed are scarce. Please provide more details on this. When exactly were the measurements performed after plasmid or siRNA transfection? If this indeed for a short while, can we truly call it an effect of LIPG on proliferation? I think we might need more tiem for that, i.e. by knocking down LIPG by shRNA.

4. The information on statistical analysis is scarce. What kind of test was used, how large were the numbers? Moreover, the results with WY-14643 and the overexpression need to be analyzed with a two-way ANOVA but it looks like the authors performed a simple t-test.

5. Figure 1C shows data on the LIPG expression in various goat tissues but there is no information in the material and methods section on how these tissues were collected.

6. Although the authors describe in line 191-192 that there was a suppressed intracellular TAG after LIPG knockdown this is actually not the case according to Figure 2F since a don’t see significant asterisk.

7. Looking at figures 2B and 4B get the impression that the cells are still pre-adipocytes with relatively low lipid content but I cannot completely be sure since a ‘simple’ picture showing the cells in culture is missing. In addition, the data in Figures 2G and 4G show that the proliferation is very low in the control situation making the effects of LIPG knockdown on this rather questionable. Is this truly an effect?

8. Is the very small induction of the FADS1 mRNA expression shown in Figure 3B truly an effect? I doubt it with such a small difference.

9. I’m puzzled by the fact that the si-LIPG treated cells have a higher PPARa level (see Figure 7A-C) but at the same time have reduced mRNA expression of typical PPARa target genes such as CPT1a and ACOX1). Please explain in the discussion why this might linked. Or is something else going on?

10. In Figure 7D-G the authors show data suggesting rescue of the LIPG overexpression effect by treatment with the PPARa agonist WY-14643. They fail, however, to show data on the mRNA and protein expression of PPARa itself. Vice versa, the authors show an upregulation of PPARa in the siRNA treated cells, but do not follow up on this by treating the cells with, for instance, a PPARa antagonist like GW6471.

11. It would be great of the authors could provide more information on the genes that are significantly differently expressed between si-LIPG treated and control cells. Why not provide at least a list of the top-50 up- and down-regulated genes? Now I’m only getting very curious which gene is depicted in the top right in Figure 6A. It can’t be LIPG since this gene is upregulated.

12. In lines 313-314 the authors state that the LIPG knockdown is assumed to result in elongation of fatty acids. Why do they think so? Is this because of the increased Elovl6 mRNA expression? Why didn’t the authors analyze the fatty acid composition to check this?

13. In lines 316-318 the authors state that the expression of ATGL was ‘was increased extremely significantly increased by LIPG silencing’. Apart from the strange sentence, I don’t think you can call a 10% extremely! Aren’t the authors over interpreting the mRNA data here?

Reviewer #3: In this paper by Wang et al, the authors knock down and overexpress LIPG in goat preadipocytes and differentiate into adipocytes. They find that increased LIPG increases lipid deposition and cell proliferation. There are also changes in gene expression. Some of those changes are rescued by a PPAR alpha agonist. Although the observations that LIPG overexpression could increase cell proliferation and lipid accumulation were interesting, the study suffers from several issues that should be addressed.

Issues

The authors conclude that LIPG is acting through PPAR alpha, but the data don’t strongly support this. Yes, a PPAR alpha agonist normalized some of the gene expression changes, but it did not normalize increased lipid accumulation or cell proliferation (as shown in Fig. 7 D and E).

The authors show a lot of gene expression data, but these data don’t indicate a clear mechanism (for example, ATGL and HSL go different directions) and there is no follow up (western blots, mechanistic assays) so these expression data don’t provide very much insight. Moreover, there is some lack of consistency. In figure 5D, ATGL expression decreases with LIPG overexpression, but in Figure 7F, ATGL expression increases 3 fold with LIPG overexpression.

It is not clear if any of the changes seen with LIPG overexpression actually require LIPG activity. This could easily be tested with Tetrahydrolipstatin (orlistat).

For both overexpression and knockdown studies, the assays measuring actual TG levels, don’t show a difference between control cells and overexpression or knockdown cells.

In LIPG overexpression assays, LIPG expression is increased over a thousand fold. This seems to indicate that there is very little LIPG expression normally in these cells, which might suggest LIPG is not strongly relevant to adipogenesis in these cells.

Overall the writing needs editing for clarity and accuracy.

The authors need to indicate what statistical tests they used to determine p values.

6. PLOS authors have the option to publish the peer review history of their article (what does this mean? ). If published, this will include your full peer review and any attached files.

**Do you want your identity to be public for this peer review?** For information about this choice, including consent withdrawal, please see our Privacy Policy .

Reviewer #1: No

Reviewer #2: No

Reviewer #3: No

---

## [Author Response · Author response to Decision Letter 0]

5 Mar 2024

Response to Reviewers

Dear editors and reviewers:

Thanks for your comments concerning our manuscript entitled “LIPG promotes lipid deposition through PPARα signaling pathway in goat intramuscular preadipocyte” (PONE-D-23-34403). Those comments were valuable and very helpful. We have read the comments and the instructions for authors carefully. Revised portions are marked in red in the Revised Manuscript with Track Changes.

To editors

Q1: Please ensure that your manuscript meets PLOS ONE's style requirements, including those for file naming.

R1: Thank you for your comments, we have revised the manuscript as requested by PLOS ONE, including correctly named file names.

Q2: We noticed you have some minor occurrence of overlapping text with the following previous publication(s), which needs to be addressed:

- https://www.mdpi.com/1422-0067/24/17/13415

In your revision ensure you cite all your sources (including your own works), and quote or rephrase any duplicated text outside the methods section. Further consideration is dependent on these concerns being addressed.

R2: Thank you for your comments, we have redescribed the section of overlapping and correctly quote the relevant literature.

Q3: Thank you for stating the following financial disclosure: [Jiangjiang Zhu was supported by Sichuan Science and Technology Program (2021YFYZ0003)；Jiangjiang Zhu was supported by Zhejiang Science Technology Program (2022C04017)；Hua Xiang was supported by Sichuan Science and Technology Program (2022NSFC0082)；Yinggui Wang was supported by Graduate innovative scientific research project of Southwest Minzu University (ZD2022726)；].

R3: Thanks to your comments, we've added the Author Contributions and Author Contributions sections to the manuscript (line381-397).

Q4: PLOS ONE now requires that authors provide the original uncropped and unadjusted images underlying all blot or gel results reported in a submission’s figures or Supporting Information files. This policy and the journal’s other requirements for blot/gel reporting and figure preparation are described in detail at https://journals.plos.org/plosone/s/figures#loc-blot-and-gel-reporting-requirements and https://journals.plos.org/plosone/s/figures#loc-preparing-figures-from-image-files. When you submit your revised manuscript, please ensure that your figures adhere fully to these guidelines and provide the original underlying images for all blot or gel data reported in your submission. See the following link for instructions on providing the original image data: https://journals.plos.org/plosone/s/figures#loc-original-images-for-blots-and-gels.

R4: Thanks to your comments, we have added the original images of all the blot results in the supplementary material "S1_raw_images.pdf".

Q5: PLOS requires an ORCID iD for the corresponding author in Editorial Manager on papers submitted after December 6th, 2016. Please ensure that you have an ORCID iD and that it is validated in Editorial Manager. To do this, go to ‘Update my Information’ (in the upper left-hand corner of the main menu), and click on the Fetch/Validate link next to the ORCID field. This will take you to the ORCID site and allow you to create a new iD or authenticate a pre-existing iD in Editorial Manager. Please see the following video for instructions on linking an ORCID iD to your Editorial Manager account: https://www.youtube.com/watch?v=_xcclfuvtxQ"

R5: As requested, we have updated the ORID iD in Editorial Manager.

Q6: We note that Figure(s) 2B, 2D, 2G, 4B, 4D, 4G, 6B and 7D in your submission contain copyrighted images. All PLOS content is published under the Creative Commons Attribution License (CC BY 4.0), which means that the manuscript, images, and Supporting Information files will be freely available online, and any third party is permitted to access, download, copy, distribute, and use these materials in any way, even commercially, with proper attribution. For more information, see our copyright guidelines: http://journals.plos.org/plosone/s/licenses-and-copyright.

R6: All of these images were taken by the author, so there are no copyright issues.

To Reviewer #1

Q1: The manuscript is very poorly written with very poor English with a lot of grammar and syntax errors which make the reading very difficult.

R1: Thanks for your comments, we have made a lot of changes to the article in terms of language and writing.

Q1: Experimental details regarding Oil Red, BODIPY, and EDU staining are missing: the authors should provide information on the time frame between siRNA- and plasmid- transfection and staining with Oil Red, BODIPY, and EDU; were the cells incubated in a full medium with 10% FCS?

R1: Thanks for the comments. We have added a subsection to the Materials and Methods section to describe the time frame for transfection and subsequent experiments (line 113-122).

"After the end of transfection, the cell culture medium was replaced with induction medium (MEM/F12, 10% FBS, 1% Penicillin-Streptomycin Solution, 50 μmol/L oleic acid (Sigma, Tokyo, Japan))(31). After 48 hours, these cells were utilized for subsequent analyses including Oil Red O staining, BODIPY staining, triglyceride assay, EDU staining, and qPCR. "

The media used to culture the cells contained 10% FCS.

Q2: Page 6 lane 115; cells were treated with triglyceride lysate? What does this mean?

R2: We have changed the irregular description "triglyceride lysate" to the correct description "cell lysis" (line 139).

Q3: LIPG Western blot (WB) of untreated cells as well as of LIPG silenced and overexpressing cells should be performed and shown; WB of cell lysates and of heparin medium should be shown.

R3: We have assayed the protein content of LIPG in LIPG silenced and overexpressing cells using WB. These results were shown in Figure 2B-C and Figure 4B-C. (line 211, line 252)

Q4: It is not stated which statistical tests were used for the analyses.

R4: We have added a sentence in the statistical analysis section to describe the statistical tests that were used. (line 184-185)

"T-tests were used for statistical analysis when there were only two groups, and one-way ANOVA was used when there were three or more groups."

Q5: The authors should discuss why LPL is upregulated after LIPG overexpression. Previous studies have shown upregulation of EL in WAT of LPL-KO mice.

R5: Thanks for your comments, we've added this section to the discussion section. (line 352-357)

"We are puzzled that overexpression of LIPG increased LPL expression in goat intramuscular precursor adipocytes. Previous studies have shown that Lipg-/- mice have higher levels of LPL expression than WT mice, and it has been demonstrated that LIPG and LPL collaborate to promote efficient triglyceride (TG)-rich lipoproteins (TRLs) lipolysis (36). The differences between our results and the reported in mice may be due to cell-individual differences and ethnic differences between mice and goats. Obtaining knock-out goats for in vivo studies of LIPG gene function related to IMF deposition is likely not feasible due to the high cost and concern for animal welfare."

Q6: Description of Fig. 7 is confusing and something is not correct. The authors show the impact of LIPG silencing on PPARa, then the impact of overexpression on lipid accumulation (BODIPY) and also graph with EDU results, but state in the legend it is BODIPY. Fig. 7F shows upregulated ATGL after LIPG overexpression which is contradictory to Figs 3 and 5.

R6: Thank you for your recommendation, we have reorganized Figure 7 and have removed the inappropriate description.

Q7: Page 13, lane 305: It is not correct that LIPG was identified first in endothelial-type adipocytes; it was in HUVEC.

R7: Thank you very much for pointing out the inprecision in our writing. We have changed "endothelial type adipocytes" to "Human Umbilical Vein Endothelial Cells (HUVEC)". (line 339-340)

Q8: Page 14, lane 331: The reference 48 is wrong; this paper is on HL and not on LIPG and PPARa.

R8: We have corrected the references (Ahmed in drugi, 2006). (line 369)

References:

Ahmed, W., Orasanu, G., Nehra, V., Asatryan, L., Rader, D. J., Ziouzenkova, O. in Plutzky, J. (2006). High-density lipoprotein hydrolysis by endothelial lipase activates PPARalpha: a candidate mechanism for high-density lipoprotein-mediated repression of leukocyte adhesion. Circulation Reserch, 98(4), 490-498. https://doi.org/10.1161/01.RES.0000205846.46812

To Reviewer #2

Q1: What was the rationale to study the role of LIPG in goat specifically? Is it indeed as mentioned on page 3 to improve goat meat quality? I bet there will be better targets than LIPG and alternative methods to do this. Please explain in more detail.

R1: Thanks for the comments. We also believe that there will be better targets than LIPG and alternative methods to improve the quality of goat meat. We studied LIPG in order to reveal the regulatory effects of LIPG on lipid deposition in goat intramuscular preadipocytes, and these efforts will help to expand the regulatory network of intramuscular fat (IMF) and provide a theoretical basis for improving meat quality. As our previous formulation could be misleading to readers, we have changed it to "Conclusively, these findings may contribute to our understanding of the lipid metabolism regulation network and provide a theoretical basis for using genetic breeding techniques to improve the quality of goat meat". (line 70-72)

Q2: How do the results in the current paper compare to data available from, for instance, LIPG knockout mice? Do they show similar effects on adipose tissue? This should be discussed in the discussion.

R2: Thank you for your comment, we have added a section to the discussion to compare with current paper. (Line 345-346, 352-355, 367-369)

Q3: I’m completely lost in the way the authors culture the cells and performed their experiments. Firstly, how are they sure that they collected the pre-adipocytes from the big piece of muscle they collected from goats? I’d advise to show some typical pre-adipocyte markers before and directly after isolation form the muscle as well as after the differentiation.

R3: Thank you for your comment! Previous studies have described the protocol to Isolated intramuscular preadipocytes and myoblasts from dorsal longest muscle. In simple terms, cells are differentiated according to the adhesion time, the intramuscular preadipocytes attach to the wall in 0-2h, and the muscle cells attach to the wall in 2-72h(Qiu in drugi, 2018).Therefore, we collected preadipocytes by removing the culture medium and nonadherent cells at 2h. At the same time, we have added a description of how preadipocytes are collected from dorsal longest muscle in Materials and Methods. (Line 93-95)

"Previous studies have shown that preadipocytes were adhere at 0-2 h and myogenic cells at 2-74 h (30). Intramuscular preadipocytes were isolated by removing nonadherent cells from the supernatant after 2 h. "

A number of adipocyte markers have been examined in our previous studies at different times of differentiation(Tian in drugi, 2022).

References:

Qiu, K., Zhang, X., Wang, L., Jiao, N., Xu, D. in Yin, J. (2018). Protein Expression Landscape Defines the Differentiation Potential Specificity of Adipogenic and Myogenic Precursors in the Skeletal Muscle. Journal of Proteome Research, 17(11), 3853-3865. https://doi.org/10.1021/acs.jproteome.8b00530

Tian, W., Xiang, H., Li, Q., Wang, Y., Zhu, J., Lin, Y. in Jois, M. (2022). Oleic acid, independent of insulin, promotes differentiation of goat primary preadipocytes. Animal Production Science, 63(2), 113-119. https://doi.org/10.1071/an21155

Q4: Secondly, is adding oleic acid enough to get adipocytes or are the cultured cells in the end still pre-adipocytes? Looking at the figures I get the impression that these cells are still the latter. So why use this protocol and not more established ones like adding a PPARgamma agonist and insulin?

R4: Our previous results showed that the cells still had good differentiation ability after 48h induction by oleic acid. Our previous results showed that preadipocytes could self-differentiate for more than one week under natural induction, although this process may be shortened under oleic acid induction, but morphologically, it is true that preadipocytes have a fibrous morphology rather than mature adipocyte morphology. In addition, we also compared two protocols, the addition of insulin and the addition of oleic acid, and found that oleic acid induced better differentiation of goat preadipocytes (Tian in drugi, 2022).

Reference:

Tian, W., Xiang, H., Li, Q., Wang, Y., Zhu, J., Lin, Y. in Jois, M. (2022). Oleic acid, independent of insulin, promotes differentiation of goat primary preadipocytes. Animal Production Science, 63(2), 113-119. https://doi.org/10.1071/an21155

Q5: Can the authors also provide data on whether the differentiation worked by for instance showing bona fide adipogenesis as well as proliferation markers at various time-points?

R5: Our previous studies have detected lipid marker genes at different time points, indicating that intramuscular precursor adipocytes have good lipid formation ability during differentiation (Tian in drugi, 2022). We added the expression profiles of proliferation marker genes at different time points to the supplementary materials, indicating that the proliferation capacity of preadipocytes still exists in spite of the decreased proliferative ability during differentiation. In addition, we also added this part in the paper. (Line 219-220)

"In the process of inducing the intramuscular preadipocytes to differentiate into adipocytes, the proliferation capacity of the cells was decreased but still existed (S2 Fig)."

Reference:

Tian, W., Xiang, H., Li, Q., Wang, Y., Zhu, J., Lin, Y. in Jois, M. (2022). Oleic acid, independent of insulin, promotes differentiation of goat primary preadipocytes. Animal Production Science, 63(2), 113-119. https://doi.org/10.1071/an21155

Q6: Thirdly, the information on when the exact experiments were performed are scarce. Please provide more details on this. When exactly were the measurements performed after plasmid or siRNA transfection? If this indeed for a short while, can we truly call it an effect of LIPG on proliferation? I think we might need more tiem for that, i.e. by knocking down LIPG by shRNA.

R6: We added the exact times of plasmid transfection, siRNA transfection, and subsequent experiments in line 120.

We are confident that siRNA is consistently inhibiting LIPG expression within 48 h, as significant interference efficiency was detected by us at 48 h in line 210.

Therefore, we truly call it an effect of LIPG on proliferation.

Q7: The information on statistical analysis is scarce. What kind of test was used, how large were the numbers? Moreover, the results with WY-14643 and the overexpression need to be analyzed with a two-way ANOVA but it looks like the authors performed a simple t-test.

R7: Thanks to your reminder, we added descriptions of the number of biological replications and statistical methods. (line 184-187)

"T-tests were used for statistical analysis when there were only two groups, and one-way ANOVA was used when there were three or more groups. The number of all biological replicates in the experiment was three (n=3). Significance was indicated using "*" p<0.05, "**" p<0.01, "***" p<0.001, "****" p<0.0001."

Q8: Figure 1C shows data on the LIPG expression in various

---

## [Decision Letter · Decision Letter 1]

19 Apr 2024

PONE-D-23-34403R1LIPG regulates lipid deposition partly through PPARα signaling pathway in goat intramuscular preadipocytesPLOS ONE

Dear Dr. Zhu,

Thank you for submitting your manuscript to PLOS ONE. After careful consideration, we feel that it has merit but does not fully meet PLOS ONE’s publication criteria as it currently stands. Therefore, we invite you to submit a revised version of the manuscript that addresses the points raised during the review process.

**ACADEMIC EDITOR: **The authors should answer all the comments of the reviewers (first and second review). The english should be further improve. ==============================

We look forward to receiving your revised manuscript.

Kind regards,

Jérôme Robert, PhD

Academic Editor

PLOS ONE

Reviewers' comments:

Reviewer's Responses to Questions

**Comments to the Author**

1. If the authors have adequately addressed your comments raised in a previous round of review and you feel that this manuscript is now acceptable for publication, you may indicate that here to bypass the “Comments to the Author” section, enter your conflict of interest statement in the “Confidential to Editor” section, and submit your "Accept" recommendation.

Reviewer #1: (No Response)

Reviewer #3: (No Response)

2. Is the manuscript technically sound, and do the data support the conclusions?

Reviewer #1: Partly

Reviewer #3: No

3. Has the statistical analysis been performed appropriately and rigorously? 

Reviewer #1: Yes

Reviewer #3: I Don't Know

4. Have the authors made all data underlying the findings in their manuscript fully available?

Reviewer #1: Yes

Reviewer #3: Yes

5. Is the manuscript presented in an intelligible fashion and written in standard English?

Reviewer #1: No

Reviewer #3: No

6. Review Comments to the Author

Reviewer #1: The authors partially improved the manuscript.

There are still some issues as listed below:

Lane 219: ..the proliferation capacity….. (S2 Fig.). This Fig shows lipid accumulation (BODIPY) but not proliferation.

Lane 262:

Addition of orlistat did not restore… I would suggest here and in the other parts of the manuscript replacing restore with: attenuate or inhibit or prevent or decrease.

The authors did not describe in the methods and figure legend the experimental details regarding orlistat treatment (concentration, duration of pre-treatment etc.).

The authors should test whether the applied amounts/concentrations of orlistat indeed attenuate LIPG activity.

If orlistat indeed had no impact on LIPG-overexpression-induced lipid accumulation (which is surprising and very interesting) the authors should try to provide an explanation how LIPG for example by its bridging function or some other mechanisms, other than by provision of FFA, affects PPAR activity and lipid accumulation.

Lanes: 314-320:

This part should be more carefully and precisely described. It is not correct as presently presented that OE-LIPG+WY14643 groups had more lipid droplets than OE-LIPG…..

It is not correct that PPARa can partially rescue (better: decrease)…The Figure shows that PPARa agonist significantly decreased lipid accumulation to the levels in the corresponding controls.

There is a discrepancy between Figure legend and the text regarding labeling of the subfigures. (Text 7A-F; legend A-C).

Lane 355: Not ethnic, but species specific differences.

Reviewer #3: In this resubmitted manuscript by Wang et al, the authors knock down and overexpress LIPG in goat preadipocytes and differentiate into adipocytes. They find that increased LIPG increases lipid deposition and cell proliferation. There are also changes in gene expression. Some of those changes are rescued by a PPAR alpha agonist. Although the observations that LIPG overexpression could increase cell proliferation and lipid accumulation were interesting and the authors addressed some concerns, the study continues to suffer from several major issues and many previous concerns were not adequately addressed.

Issues

The authors continue to assert that LIPG regulates cell proliferation and lipid deposition at least in part through PPAR alpha, but the data still don’t strongly support this. Yes, a PPAR alpha agonist normalized some of the gene expression changes, but data for cell proliferation is not shown and the agonist did not normalize increased lipid accumulation. True total lipid accumulation is lower with the agonist but it is lower both with and without LIPG overexpression and the induction by LIPG overexpression is similar suggesting that the change in lipid accumulation is independent of PPAR.

In the original submission, the authors do not perform any experiments to address the mechanism whereby endothelial lipase may induce lipid accumulation. In the revised experiment the authors perform an experiment with orlistat, which should inhibit the enzymatic activity of endothelial lipase. Unfortunately the authors do not perform controls to show that the orlistat was effective (experiments demonstrating a change in lipase activity with LIPG overexpression and with orlistat treatment should be included). Moreover if the orlistat was effective the authors show that there is no change in lipid accumulation (cell proliferation was not tested). This would suggest that the effect on lipid accumulation is not dependent on the enzymatic activity of endothelial lipase. The authors state this, at yet their hypothesis at the conclusion of their discussion is that EL-mediated hydrolysis liberates fatty acids that act on PPAR. Lipolysis requires endothelial lipase activity so how could this happen if endothelial lipase activity is not required?

As before, the authors show a lot of gene expression data, but these data don’t indicate a clear mechanism (for example, ATGL and HSL go different directions) and there is no follow up (western blots, mechanistic assays) so these expression data don’t provide very much insight. Moreover, to address the lack of consistency with the ATGL expression data in the original submission, they have simply removed the ATGL expression data from figure 7 without adequate explanation.

For both overexpression and knockdown studies, the assays measuring actual TG levels, don’t show a difference between control cells and overexpression or knockdown cells. There may be a non-statistically significant trend, but the data does not match the BODIPY or Oil Red O data.

Graphs should show individual points not just bars and error bars.

Overall the writing continues to suffer from a lack of clarity and accuracy. For example:

-The last sentence of the abstract is repeated.

-When the authors perform experiments with PPAR agonists they often use the terms “restore” or “rescued” when I think what they mean is “reversed”.

-Authors mention that LIPG was identified at major differential gene between fat-tailed and thin-railed sheep. Do they mean thin-tailed, and though they say they have different lipid storage patterns they don’t explain what those are.

-The authors state, “Our findings are similar to previous reports that interfered LIPG inhibits lipid deposition and cell proliferation in goat intramuscular preadipocytes.” But none of the references that follow (41, 42, and 43) examined LIPG or used goat intramuscular preadipocytes.

7. PLOS authors have the option to publish the peer review history of their article (what does this mean? ). If published, this will include your full peer review and any attached files.

**Do you want your identity to be public for this peer review?** For information about this choice, including consent withdrawal, please see our Privacy Policy .

Reviewer #1: No

Reviewer #3: No

---

## [Author Response · Author response to Decision Letter 1]

18 Nov 2024

Dear editors and reviewers:

Thanks for your comments concerning our manuscript entitled “LIPG promotes lipid deposition through PPARα signaling pathway in goat intramuscular preadipocyte” (PONE-D-23-34403). Those comments were valuable and very helpful. We have read the comments and the instructions for authors carefully. Revised portions are marked in red in the Revised Manuscript with Track Changes.

To Reviewer 1

Q1: Lane 219: ..the proliferation capacity.. (S2 Fig.). This Fig shows lipid accumulation (BODIPY) but not proliferation.

R1: Thank you for pointing out this discrepancy. We have corrected the text to accurately reflect the content of the figure. In the process of inducing the intramuscular preadipocytes to differentiate into adipocytes, the proliferation capacity of the cells was decreased by EDU staining (Fig 2I-2J).” We appreciate your attention to detail.

Q2: Lane 262:Addition of orlistat did not restore… I would suggest here and in the other parts of the manuscript replacing restore with: attenuate or inhibit or prevent or decrease.

R2: Thank you for your helpful suggestion. We have replaced "restore" with "attenuate" in the revised sentence, which now reads: " Addition of orlistat did not attenuate the increase in lipid deposition caused by overexpression of LIPG (S2 Fig)".

Q3: The authors did not describe in the methods and figure legend the experimental details regarding orlistat treatment (concentration, duration of pre-treatment etc.).

R3: Thank you for pointing this out. We have added the relevant experimental details regarding the orlistat treatment to the Materials and Methods section. The revised text now reads: " Transfected cells were treated with 20 μM orlistat (dissolved in DMSO) and cultured for an additional 48 h." We hope this clarifies the methodology used in our experiments.

Q4: The authors should test whether the applied amounts/concentrations of orlistat indeed attenuate LIPG activity. If orlistat indeed had no impact on LIPG-overexpression-induced lipid accumulation (which is surprising and very interesting) the authors should try to provide an explanation how LIPG for example by its bridging function or some other mechanisms, other than by provision of FFA, affects PPAR activity and lipid accumulation.

R4: Thank you for this valuable recommendation. We understand the importance of confirming whether the applied concentration of orlistat attenuates LIPG activity. However, due to current limitations in our laboratory facilities, we are unable to conduct this specific experiment at this time. We have, however, added a discussion of this limitation and its potential implications in the Discussion section to acknowledge the need for future investigation:

"However, our data from the orlistat experiment suggest that lipid accumulation changes associated with LIPG overexpression may not be solely dependent on its enzymatic activity, as orlistat treatment did not reverse the increased lipid accumulation. This raises the possibility that LIPG could also regulate lipid deposition through non-enzymatic mechanisms, particularly in goat intramuscular preadipocytes. While these findings contribute to our understanding, further experimental validation is needed to clarify the precise mechanisms—both enzymatic and potential non-enzymatic—through which LIPG regulates lipid metabolism and the PPARα pathway."

We hope this addresses your concern and we look forward to future investigations on this topic.

Q5: This part should be more carefully and precisely described. It is not correct as presently presented that OE-LIPG+WY14643 groups had more lipid droplets than OE-LIPG…..

It is not correct that PPARa can partially rescue (better: decrease)…The Figure shows that PPARa agonist significantly decreased lipid accumulation to the levels in the corresponding controls.

R5: Thank you very much for your careful review and valuable comments. We apologize for the errors and any confusion caused by these inaccuracies in the original manuscript. We have now thoroughly revised the relevant sections to ensure clarity and accuracy.

We appreciate your pointing out the inaccurate description of lipid droplet accumulation in the OE-LIPG+WY14643 groups. We have corrected this in the revised manuscript to more precisely reflect that the PPARα agonist significantly decreased lipid accumulation to levels similar to the corresponding controls. (Line 319-320)

Furthermore, to provide a more accurate description, we have amended the phrasing regarding the “partially rescue” effect of PPARα. Instead, we now describe that the PPARα agonist significantly decreased lipid accumulation, not partially rescuing the effect.To better reflect these changes and the focus of our findings, we have also modified the title of the manuscript to: “LIPG-mediated regulation of lipid deposition and proliferation in goat intramuscular preadipocytes involves the PPARα signaling pathway.”Thank you again for your constructive feedback.

Q6: There is a discrepancy between Figure legend and the text regarding labeling of the subfigures. (Text 7A-F; legend A-C).

R6: Thank you for pointing out the discrepancy between the figure legend and the text regarding the labeling of the subfigures. We have carefully reviewed both the figure and the legend and made the necessary corrections. Specifically, we have updated the legend to label the subfigures as 7A-7F, which now matches the description in the text. We believe these changes address the issue and enhance the accuracy of the manuscript. Thank you again for your valuable feedback.

Q7: Lane 355: Not ethnic, but species specific differences.

R7: Thank you for your valuable comment. We have corrected the text to reflect that the differences observed are not ethnic, but rather species-specific.

To Reviewer 1

Q1: The authors continue to assert that LIPG regulates cell proliferation and lipid deposition at least in part through PPAR alpha, but the data still don’t strongly support this. Yes, a PPAR alpha agonist normalized some of the gene expression changes, but data for cell proliferation is not shown and the agonist did not normalize increased lipid accumulation. True total lipid accumulation is lower with the agonist but it is lower both with and without LIPG overexpression and the induction by LIPG overexpression is similar suggesting that the change in lipid accumulation is independent of PPAR.

R1: Thank you very much for your insightful comments, which have prompted us to reconsider the scope of our title and clarify the role of the PPARα signaling pathway in LIPG-mediated regulation. In response to your feedback, we have adjusted the title to better reflect our findings that, while PPARα signaling is involved, it does not fully account for the effects of LIPG on lipid deposition and cell proliferation in goat intramuscular preadipocytes. The revised title is as follows:

“LIPG-mediated regulation of lipid deposition and proliferation in goat intramuscular preadipocytes involves the PPARα signaling pathway”

This updated title emphasizes that PPARα signaling is one pathway among potentially several through which LIPG influences lipid metabolism and cell proliferation, aligning more closely with our data. We acknowledge that while the PPARα agonist (WY14643) reduced lipid accumulation to some extent in the LIPG-overexpressing cells, it did not fully normalize the increase in lipid deposition, suggesting that additional signaling mechanisms may also contribute to LIPG’s regulatory role.

Q2: In the original submission, the authors do not perform any experiments to address the mechanism whereby endothelial lipase may induce lipid accumulation. In the revised experiment the authors perform an experiment with orlistat, which should inhibit the enzymatic activity of endothelial lipase. Unfortunately the authors do not perform controls to show that the orlistat was effective (experiments demonstrating a change in lipase activity with LIPG overexpression and with orlistat treatment should be included). Moreover if the orlistat was effective the authors show that there is no change in lipid accumulation (cell proliferation was not tested). This would suggest that the effect on lipid accumulation is not dependent on the enzymatic activity of endothelial lipase. The authors state this, at yet their hypothesis at the conclusion of their discussion is that EL-mediated hydrolysis liberates fatty acids that act on PPAR. Lipolysis requires endothelial lipase activity so how could this happen if endothelial lipase activity is not required?

R2: Thank you for your insightful and constructive feedback. We greatly appreciate your careful analysis of our study, particularly your comments regarding the orlistat experiment and its implications for LIPG's enzymatic activity. In response to your concerns, we acknowledge that orlistat treatment did not reverse the increase in lipid accumulation induced by LIPG overexpression. This observation suggests that the regulation of lipid accumulation by LIPG may not be entirely dependent on its enzymatic activity. As we discussed in the revised manuscript, we now hypothesize that LIPG might also exert its regulatory effects through non-enzymatic mechanisms in addition to its known enzymatic function. This aspect of LIPG’s role requires further investigation, and we plan to explore these non-enzymatic pathways in future studies.

We also recognize that our study did not provide direct evidence to confirm that orlistat effectively inhibited LIPG enzymatic activity. We agree with your suggestion that future experiments should include controls demonstrating a change in lipase activity with LIPG overexpression and orlistat treatment. This would strengthen our conclusions and provide clearer evidence regarding the enzymatic versus non-enzymatic contributions of LIPG. Despite these limitations, our current findings suggest that LIPG regulates lipid deposition through multiple pathways, potentially including both enzymatic and non-enzymatic mechanisms. We have emphasized this point in the revised discussion, acknowledging the need for further experimental validation. Once again, thank you for your valuable input, which has helped us clarify our hypotheses and the direction of future research. We are committed to addressing these issues in subsequent experiments.

Q3: As before, the authors show a lot of gene expression data, but these data don’t indicate a clear mechanism (for example, ATGL and HSL go different directions) and there is no follow up (western blots, mechanistic assays) so these expression data don’t provide very much insight. Moreover, to address the lack of consistency with the ATGL expression data in the original submission, they have simply removed the ATGL expression data from figure 7 without adequate explanation.

R3: Thank you for your valuable feedback. We appreciate your concerns regarding the gene expression data and the lack of mechanistic follow-up, as well as the issue with the ATGL expression data in Figure 7. In response to your first point, we understand that while gene expression data provide valuable insights into the transcriptional regulation of lipid metabolism, they alone do not fully elucidate the underlying mechanisms. We agree that further validation through Western blot analysis and mechanistic assays would significantly strengthen the conclusions of our study. Unfortunately, due to experimental constraints, we were unable to perform these additional experiments in the current version of the manuscript. However, we acknowledge the importance of these follow-up experiments and plan to incorporate them in future studies to provide a more comprehensive understanding of LIPG’s role in lipid accumulation and its impact on gene regulation. Regarding the issue with ATGL expression data, we recognize that the inconsistency in the ATGL expression data between our original submission and the revised manuscript may have raised concerns. In the revised manuscript, we made the decision to remove the ATGL data from Figure 7 due to its lack of consistency with the other experimental results and to avoid any confusion. We should have provided a more thorough explanation for this change in the manuscript. To address this, we have revised the text in the discussion to clarify that the ATGL data were removed due to inconsistencies with the overall findings and we acknowledge the need for further investigation into its potential role in lipid metabolism in the context of LIPG regulation. We believe that the data we have presented, along with the modifications in the manuscript, still provide valuable insights into the role of LIPG in regulating lipid deposition through its effects on various lipid metabolism-related genes and the PPARα signaling pathway. We hope that future experiments, including Western blotting and other mechanistic assays, will help to clarify these findings in more detail. Thank you again for your thoughtful suggestions. We will take these points into consideration for the next phase of our research and in preparing future revisions of the manuscript

Q4: For both overexpression and knockdown studies, the assays measuring actual TG levels, don’t show a difference between control cells and overexpression or knockdown cells. There may be a non-statistically significant trend, but the data does not match the BODIPY or Oil Red O data.

R4: We appreciate your thoughtful review of our manuscript and your comments regarding the discrepancy between the TG levels and the BODIPY and Oil Red O data.

We would like to clarify that, although the TG levels did not show statistically significant differences between control, overexpression, and knockdown cells, the trend observed was consistent with the BODIPY and Oil Red O staining results. Specifically, overexpression of LIPG led to an increase in TG levels, while knockdown resulted in a decrease, mirroring the pattern observed in lipid accumulation with BODIPY and Oil Red O staining. This consistency suggests that, despite the lack of statistical significance in the TG assay, the trend supports our findings regarding the regulation of lipid deposition by LIPG. We hope this explanation helps address the concerns raised, and we appreciate your constructive input as it has guided us in clarifying this point in our revised manuscript.

Q5: Graphs should show individual points not just bars and error bars.

R5: Thank you for your valuable feedback. In response to your suggestion, we have revised the graphs to include individual data points alongside the bars and error bars. This adjustment allows for a clearer representation of data distribution and variability, providing a more detailed view for the readers. We hope this modification meets your expectations and enhances the clarity and quality of our presentation.

Q6: The last sentence of the abstract is repeated.

R6: Thank you for pointing this out. We have revised the abstract to eliminate the repetition in the last sentence. The revised version now reads more clearly and concisely. We appreciate your attention to this detail.

Q7: When the authors perform experiments with PPAR agonists they often use the terms “restore” or “rescued” when I think what they mean is “reversed”.

R7: Thank you for your valuable comment. We appreciate your suggestion. Upon review, we agree that the terms “restore” or “rescued” may not be the most appropriate in this context. We have revised the manuscript to use the term "reversed" where applicable, to better reflect the intended meaning of the experiments with PPAR agonists. We believe this change clarifies the results more accurately. Thank you again for helping improve the clarity of our manuscript.

Q8: Authors mention that LIPG was identified at major differential gene between fat-tailed and thin-railed sheep. Do they mean thin-tailed, and though they say they have different lipid storage patterns they don’t explain what those are.

R8: Thank you for your comment. We apologize for the confusion caused by the incorrect spelling and have corrected it to "thin-tailed" as you suggested. We acknowledge that the reference cited (27) does not explicitly describe the specific lipid storage patterns in fat-tailed and thin-tailed sheep. T

---

## [Editor Report · Decision Letter 2]

7 Jan 2025

LIPG-mediated regulation of lipid deposition and proliferation in goat intramuscular preadipocytes involves the PPARα signaling pathway

PONE-D-23-34403R2

Dear Dr. Zhu,

We’re pleased to inform you that your manuscript has been judged scientifically suitable for publication and will be formally accepted for publication once it meets all outstanding technical requirements.

Kind regards,

Jérôme Robert, PhD

Academic Editor

PLOS ONE
---

## [Editor Report · Acceptance letter]

PONE-D-23-34403R2

PLOS ONE

Dear Dr. Zhu,

I'm pleased to inform you that your manuscript has been deemed suitable for publication in PLOS ONE. Congratulations! Your manuscript is now being handed over to our production team.

Kind regards,

on behalf of

Dr. Jérôme Robert

Academic Editor

PLOS ONE